# Dynamic Neural Response Tuning

**Tian Qiu, Wenxiang Xu, Lin Chen, Linyun Zhou, Zunlei Feng\*, Mingli Song**
Zhejiang University
{tqiu,xuwx1996,lin_chen,zhoulyaxx,zunleifeng,songml}@zju.edu.cn

## Abstract

Artificial Neural Networks (ANNs) have gained widespread applications across various areas in recent years. The ANN design was initially inspired by principles of biology. The biological neural network's fundamental response process comprises information transmission and aggregation. The information transmission in biological neurons is often achieved by triggering action potentials that propagate through axons. ANNs utilize activation mechanisms to simulate such biological behavior. However, previous studies have only considered static response conditions, while the biological neuron's response conditions are typically dynamic, depending on multiple factors such as neuronal properties and the real-time environment. Therefore, the dynamic response conditions of biological neurons could help improve the static ones of existing activations in ANNs. Additionally, the biological neuron's aggregated response exhibits high specificity for different categories, allowing the nervous system to differentiate and identify objects. Inspired by these biological patterns, we propose a novel Dynamic Neural Response Tuning (DNRT) mechanism, which aligns the response patterns of ANNs with those of biological neurons. DNRT comprises Response-Adaptive Activation (RAA) and Aggregated Response Regularization (ARR), mimicking the biological neuron's information transmission and aggregation behaviors. RAA dynamically adjusts the response condition based on the characteristics and strength of the input signal. ARR is devised to enhance the network's ability to learn category specificity by imposing constraints on the network's response distribution. Extensive experimental studies indicate that the proposed DNRT is highly interpretable, applicable to various mainstream network architectures, and can achieve remarkable performance compared with existing neural response mechanisms in multiple tasks and domains. Code is available at https://github.com/horrible-dong/DNRT.

## 1 Introduction

In recent years, Artificial Neural Networks (ANNs) (McCulloch & Pitts, 1943; Jain et al., 1996; LeCun et al., 2015; Abiodun et al., 2018) have made significant milestones in various computer vision tasks (Voulodimos et al., 2018), such as image classification, as well as numerous other tasks and domains (Wu et al., 2020; Zhang et al., 2023).

The idea of ANNs was initially derived from the perception pattern of the human brain (McCulloch & Pitts, 1943). Multilayer Perceptron (MLP) (McClelland et al., 1987), a classical ANN model that uses neurons as its basic unit, resembles that of biological neural networks and led to significant breakthroughs. Subsequently, Convolutional Neural Networks (CNNs) (LeCun et al., 1998) were developed to simulate local visual perception. Today, the advent of abundant computing power has led to further progress in the popular Vision Transformers (ViTs) (Han et al., 2022) that integrate attention mechanisms (Vaswani et al., 2017) and MLP structures (McClelland et al., 1987).

The above ANNs utilize activation mechanisms (Dubey et al., 2022) that simulate the behavior of biological neurons transmitting the information. When the external signal reaches a specific intensity, the activation mechanism transmits the signal to the next layer; otherwise, in most cases, it suppresses the transmission of the signal. Rectified Linear Unit (ReLU) (Glorot et al., 2011) is a commonly used activation mechanism in deep learning, with a fixed response threshold of zero, allowing only

---

\*Corresponding author.

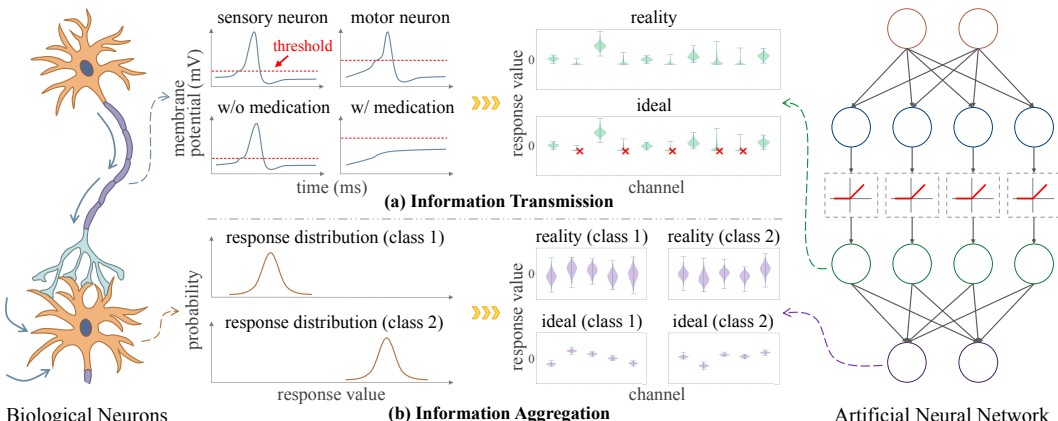

Figure 1: Observations on biological neurons and the way they have inspired the proposed Dynamic Neural Response Tuning (DNRT). The observations include two aspects of information processing: information transmission and information aggregation. (a) The threshold membrane potential required for transmitting information through neurons depends on the characteristics of neurons and the real-time environment due to its dynamically changing nature. (b) The information aggregated by neurons is highly category-specific. The proposed DNRT aims to align the response patterns of ANNs with those of biological neurons, thus addressing the ANN's corresponding observed imperfections.

inputs greater than zero to pass while other inputs are set to zero. Another classical activation form is Gaussian Error Linear Unit (GELU) (Hendrycks & Gimpel, 2016), typically used in Transformers. It suggests that the information transmission depends on the fixed priori input distribution, i.e., an input value ranked later among all inputs is more likely to be suppressed, whose curve is also predefined, similar to ReLU but better simulating real-world scenarios.

However, as shown in Figure 1(a), neuroscience research observed that the threshold membrane potential depends on neuronal properties and the real-time environment due to its dynamically changing nature (Kandel et al., 1991; Fontaine et al., 2014; Azouz & Gray, 2000; Sun, 2009; Arancio et al., 1995; Knowland et al., 2017). Similarly, in ANNs, different neurons have different functions, and the strength of inputs for different objects or categories also varies. Therefore, the input strength cannot fully indicate the relevance of the input information. Considering these issues, we believe that the dynamic response pattern of biological neurons can be applied to improve the static response pattern of the existing activation mechanisms in ANNs.

Additionally, as shown in Figure 1(b), after the signals aggregate in the biological neuron, the response distributions exhibit significant differences across categories, called the response specificity of neurons (Kreiman et al., 2000). In experiments with ANNs, we have observed similar phenomena. By training different categories of samples, the ANN can learn specificity similar to that of neurons. Furthermore, ANNs with poor performance tend to present a Gaussian distribution with higher variance under the stimuli from a specific category. The Gaussian distribution is a common assumption (Fischer, 2011) that reflects the presence of error in ANN's responses, and the distribution variance reflects the magnitude of this error. So, the high Gaussian variances indicate the specificity of the response is not strong enough, making it difficult to distinguish between different categories through their responses, posing a significant challenge for the ANN to make correct judgments.

This paper explores the application of biological neural response patterns in transmitting and aggregating information within various ANN architectures. A novel mechanism called Dynamic Neural Response Tuning (DNRT) is proposed. DNRT comprises Response-Adaptive Activation (RAA) and Aggregated Response Regularization (ARR), mimicking the biological information transmission and aggregation. RAA dynamically adjusts the response conditions based on the characteristics and strength of input signals, simulating the corresponding behavior of biological neurons. ARR imposes constraints on the distribution of aggregated neural responses based on historical statistics on category signals, enhancing the neural networks' ability to learn category specificity. Experiments on mainstream visual classification networks indicate that the proposed DNRT outperforms existing neural response mechanisms in enhancing classification accuracy and helps to differentiate the neural

response under different categories of stimuli. Additionally, DNRT exhibits the capacity to generalize effectively to other tasks and domains, showcasing its remarkable versatility.

The contribution of this paper is the simulation of dynamic biological neural response patterns through Response-Adaptive Activation and Aggregated Response Regularization to address corresponding observed imperfections in ANNs. The study provides new concepts and techniques for constructing ANNs and valuable insights into in-depth neural information processing research. Extensive experiments across various mainstream ANN architectures, tasks, and domains indicate that the proposed method surpasses existing neural response mechanisms, delivering outstanding performance.

## 2 RELATED WORK

### 2.1 NEURAL RESPONSE IN NEUROSCIENCE

Neural response refers to the manner in which the nervous system reacts to stimuli from external sources (Kandel et al., 2000). The two primary processes involved in neural response are information transmission and aggregation (Kandel et al., 1991).

**Information transmission.** Through the axons of biological neurons, signals are transmitted to the neuron's terminal, and neurotransmitters are released. Then, the signal is transmitted to other neurons through synapses (Kandel et al., 2000). The information transmission depends on the response condition. For spiking neurons, when the stimulus surpasses the threshold, an action potential that travels to the terminal is produced, resulting in the release of neurotransmitters (Hodgkin & Huxley, 1952). The neuron will not generate an action potential if the stimulus is below the threshold. Neurons with different functions possess differing response thresholds (Kandel et al., 1991). The response threshold of neurons can also be influenced by extrinsic elements such as medications (Arancio et al., 1995) and nervous system ailments (Knowland et al., 2017). Moreover, the dynamic threshold is a regulatory mechanism of biological neurons, which has been widely observed in various nervous systems (Fontaine et al., 2014; Azouz & Gray, 2000; Sun, 2009).

**Information aggregation.** Biological neurons receive impulses from multiple other neurons and integrate these signals. Signal aggregation can occur at various levels, from local to global, reflecting the cerebral cortex's highly complex interconnectivity between neurons (Kandel et al., 1991). These aggregation processes form the neural network responsible for representing complex emotions, thoughts, and behaviors in a high-dimensional space based on the information received from this network (Kandel et al., 1991). Stimuli of the same category usually generate similar responses, and for different categories of stimuli, the information aggregated by biological neurons exhibits significant differences (Kreiman et al., 2000), indicating the response specificity of neurons.

### 2.2 ACTIVATION MECHANISM

The activation mechanism (Dubey et al., 2022) plays a pivotal role in ANNs as it defines how neurons respond to input signals and transforms them into output signals. Each activation varies in mathematical properties and nonlinear forms, allowing neurons to simulate different neural phenomena, including excitation, inhibition, and modulation. According to the prevailing activation response rule, irrelevant features are expected to be suppressed, while relevant features gain amplified influence, which can achieve data sparsity, diminish redundant information, and enable better feature distinction. Mathematically, activation mechanisms are categorized into different types, including logistic Sigmoid and Tanh variants (Xu et al., 2016; LeCun et al., 1998), Rectified Linear Unit variants (Glorot et al., 2011; Maas et al., 2013; He et al., 2015; Xu et al., 2015; Jiang et al., 2018; Chen et al., 2020), Exponential Linear Unit variants (Clevert et al., 2015; Klambauer et al., 2017; Trottier et al., 2017), Softplus variants (Dugas et al., 2000; Misra, 2019), probabilistic variants (Hendrycks & Gimpel, 2016; Xu et al., 2015; Jiang et al., 2018), multi-dimensional normalization nonlinearities (Ballé et al., 2015) and others (Maass, 1997; Wang et al., 2022).

The activations ELU (Clevert et al., 2015), SELU (Klambauer et al., 2017), and SiLU (Ramachandran et al., 2017) mentioned above, and others (Liu et al., 2020), contain learnable parameters that can adapt to various data distributions, avoiding gradient vanishing and explosion, thereby enhancing the convergence speed and precision of ANNs. However, these parameters can only regulate response strength and do not influence response-triggering conditions. The threshold of spiking biological

neurons can dynamically adjust based on environmental conditions (Fontaine et al., 2014; Azouz & Gray, 2000; Sun, 2009; Arancio et al., 1995; Knowland et al., 2017). This attribute is absent in previous ANNs' activation mechanisms, indicating the need for improvement.

## 3  OBSERVATIONS

This section explores the observed phenomena in biological neurons and ANNs. The phenomena with the biological neurons' working patterns will help fix ANNs' corresponding observed imperfections.

### 3.1  OBSERVATION 1: INFORMATION TRANSMISSION RESPONSE

As illustrated in Figure 1(a), biological neurons exhibit dynamic responses. Their response condition, i.e., the threshold membrane potential, which determines the transmission of information between neurons, depends on various factors, including the type of neurons and their internal and external context. In ANNs, samples of a given category are fed to record the distributions of its responses upon passing through an activation with a fixed response threshold, in which the horizontal coordinate is the element's (channel's) index within the feature vector, the vertical coordinate is the response value, and the area indicates the response density. We observe that some channels' response distributions are truncated. These channels and their inputs are generally considered irrelevant. Most inputs fall below the pre-set response condition (the threshold) and correctly get suppressed; however, some inputs in these channels still meet the condition and mistakenly trigger activation responses because the response condition for activation is pre-set and fixed according to prior knowledge. These incorrect responses may interfere with the network's final decision. *The dynamic response conditions in biological neurons inspire us to improve the existing activation mechanisms in ANNs.*

### 3.2  OBSERVATION 2: INFORMATION AGGREGATION RESPONSE

As illustrated in Figure 1(b), for different categories of stimuli, the information aggregated by biological neurons exhibits significant differences, indicating the response specificity of neurons. Regarding the working pattern in ANNs, it aggregates the activated signals transmitted by all neurons to produce a response output. The aggregated response distributions are captured after feeding samples of the same category. The observation indicates that the distributions are approximately Gaussian-shaped at each channel. However, the variances of these Gaussian distributions are relatively large. We usually assume a Gaussian distribution to represent the presence of error in ANN's responses. The variance of this distribution corresponds to the magnitude of the error. So, the high Gaussian variances indicate the response is pretty noisy and overlaps between categories, implying that the specificity differences between categories are not significant. As a result, the ANN is more likely to misrecognize these categories, requiring further optimization. *The class-specific aggregated responses of biological neurons inspire us to devise techniques to enhance the aggregated ones of ANNs.*

## 4  DYNAMIC NEURAL RESPONSE TUNING

### 4.1  RESPONSE-ADAPTIVE ACTIVATION

Inspired by the dynamic response conditions observed in biological neurons, a general adaptive response mechanism is devised to replace the static ones in current ANN activations.

Define an activation $A$ that operates under static response conditions. Taking GELU (Hendrycks & Gimpel, 2016) as an example of $A$, each input $x$ has a probability of $\Phi(x)$ to be activated and a probability of $(1 - \Phi(x))$ to be restrained (set to zero), and the activation probability $\Phi(x)$ rises as the input $x$ gets larger. The $A$'s output w.r.t. the input $x$ can be expressed in a form of mathematical expectation:

$$
\begin{aligned}
A(x) &= \Phi(x) \cdot 1x + (1 - \Phi(x)) \cdot 0x \\
&= x \cdot \Phi(x).
\end{aligned}
\tag{1}
$$

Here, the probability $\Phi(x) = P(X \leq x)$ is the cumulative distribution function of the standard normal distribution $X \sim \mathcal{N}(0, 1)$ (Hendrycks & Gimpel, 2016). Since the curve of $\Phi(x)$ is static/fixed, any input to $A$ will consistently follow the same response condition, potentially leading to the

phenomenon described in Observation 1 (§3.1), where some irrelevant signals may incorrectly trigger an activation response and have a detrimental impact on the network's final decision.

To address this issue, we propose a Response-Adaptive Activation (RAA) mechanism that allows the original activation to adjust its response condition according to different external inputs.

Assume that an irrelevant input $x^u$ is going to incorrectly trigger a high response $x^u \cdot \Phi(x^u)$ under the original response condition of $A$. In accordance with the intended meaning of $A$, with $x^u$ determined, it is necessary to minimize the activation probability $\Phi(x^u)$ as much as possible. To accomplish this, an offset $\Delta x^u$ can be imposed on $x^u$. Since the $\Phi$ is monotonic increasing, in this case, $\Delta x^u$ should be negative, then obtain a new response $x^u \cdot \Phi(x^u + \Delta x^u)$, which is lower than the original response $x^u \cdot \Phi(x^u)$ and suppresses the irrelevant input $x^u$. On the other hand, when a relevant input $x^r$ is going to be incorrectly suppressed under the original response condition of $A$, a positive offset $\Delta x^r$ should be applied. Hence, the issue lies in the calculation of $\Delta x$.

A straightforward solution is directly treating the offset $\Delta x$ as a learnable parameter $\alpha$. Denote the Response-Adaptive Activation as $RAA$, which can be expressed as follows:

$$RAA(x) = x \cdot \Phi(x + \alpha). \tag{2}$$

Even though this technique offers adaptive response conditioning, the activation mechanism's response condition is still somewhat inflexible since $\alpha$ uniformly influences each input.

A more dynamic alternative is determining the offset $\Delta x$ based on the input $x$ itself by introducing a mapping $f$ from the input to its desired offset. Then, the second (not the final) version of $RAA$ can be expressed as follows:

$$\begin{aligned} RAA(x) &= x \cdot \Phi(x + f(x)) \\ &= x \cdot \Phi(x + (wx + b)), \end{aligned} \tag{3}$$

where the learnable scalars $w$ and $b$ represent the weight and bias of the linear mapping $f$. They are initialized to zero to ensure an initial offset of zero. Now, $RAA$ allows the activation response condition to adapt dynamically to different inputs, which is crucial for achieving better performance.

However, *we will not adopt Eq. 3* since such element-wise operation requires too many parameters and computations. In order to efficiently assess signal relevance with minimal computational costs, we further refine the RAA's granularity to the feature vector level. Generally, neural networks extract features from input data, usually presented as a feature sequence $\{\mathbf{x}_i\}_{i=1}^{L}$ of length $L$, where each feature $\mathbf{x}_i = [x_1, x_2, ..., x_d]^{\mathrm{T}}$ is a $d$-dimensional vector. Consequently, *the final version* of $RAA$ can be expressed as follows:

$$\begin{aligned} RAA(\mathbf{x}) &= \mathbf{x} \cdot \Phi(\mathbf{x} + f(\mathbf{x})) \\ &= \mathbf{x} \cdot \Phi(\mathbf{x} + (\mathbf{w}^{\mathrm{T}}\mathbf{x} + b)), \end{aligned} \tag{4}$$

where the $d$-dimensional mapping vector $\mathbf{w} = [w_1, w_2, ..., w_d]^{\mathrm{T}}$ and scalar bias $b$ are learnable parameters used in the model, which are also initialized with zero to ensure an initial offset of zero.

The basic concept and thought of RAA hold broad applicability. With negligible parameters and computations introduced, RAA gains the ability to respond to input signals in an adaptive manner. Furthermore, this behavior is not limited to a specific context and can also be extended to other static activation forms such as ReLU (Glorot et al., 2011) etc.

## 4.2 AGGREGATED RESPONSE REGULARIZATION

According to Observation 2 (§3.2), the biological aggregated responses present high category specificity. For stimuli of a given category, the ANN's aggregated response exhibits a Gaussian distribution in each channel but with a high degree of variance. A more focused response can enhance feature specificity, thus leading the network to make more accurate decisions between categories.

To address this issue, we propose a statistics-based regularization technique called Aggregated Response Regularization (ARR). Let $K$ denote the total number of categories of the given dataset. For input data of each category $k \in \{1, 2, ..., K\}$, the historical mean/average aggregated responses $\boldsymbol{\mu}_k$ within the network are tracked separately and in real time. To avoid a rise in space complexity, the historical mean is updated by using a moving mean technique as follows:

$$\boldsymbol{\mu}_k^t = (1 - m) \cdot \boldsymbol{\mu}_k^{t-1} + m \cdot \mathbf{x}_k^t, \tag{5}$$

where the vector $\mathbf{x}_k^t = [x_1, x_2, ..., x_d]^{\mathrm{T}}$ is the network's aggregated response at time $t$ when inputting a sample from category $k$, and $d$ is the vector's dimension. $\boldsymbol{\mu}_k^{t-1}$ and $\boldsymbol{\mu}_k^t$ represent the historical mean response vectors of category $k$ at times $t-1$ and $t$, respectively. The hyperparameter $m$ is the momentum for updating the moving mean. Therefore, only $K$ vectors $\{\boldsymbol{\mu}_k\}_{k=1}^K$ need to be maintained throughout the entire process, whose memory overhead is negligible.

Next, the network's aggregated response can be made more concentrated by utilizing the historical mean. To achieve this, at each time $t$, a loss constraint $\mathcal{L}_{arr}$ is applied between the aggregated response $\mathbf{x}_k^t$ and its historical mean $\boldsymbol{\mu}_k^{t-1}$ as follows:

$$\mathcal{L}_{arr} = \frac{\|\mathbf{x}_k^t - \boldsymbol{\mu}_k^{t-1}\|_1}{d}, \tag{6}$$

where $d$ is the dimension of the vector $\mathbf{x}$ and $\boldsymbol{\mu}$. Constraining the new aggregated response to be closer to the historical mean is equivalent to reducing the variance of the Gaussian distribution.

A deep ANN includes multiple layers, and the aggregated response after the activation in each layer is capable of calculating the mean value of historical aggregated responses and applying regularization to new ones. Therefore, the final loss $\mathcal{L}$ can be expressed as

$$\mathcal{L} = \mathcal{L}_{task} + \lambda \cdot \frac{1}{J} \sum_{j=1}^J \mathcal{L}_{arr}^j, \tag{7}$$

where $\mathcal{L}_{task}$ is the primary loss for the specific task; for example, in the context of a standard classification task, $\mathcal{L}_{task}$ represents the cross-entropy loss. $J$ is the number of layers in the network that have ARR applied, and $\lambda$ is the balanced parameter. Introducing this new loss function helps the network to enhance category response specificity. Especially when dealing with small datasets, the proposed ARR can better leverage limited data and avoid the risk of overfitting. Moreover, as a training strategy, the proposed approach does not affect the inference speed.

### 4.3 Neural Network Learning

The proposed Response-Adaptive Activation (RAA) mechanism can replace the network's original ones, and the proposed Aggregated Response Regularization (ARR) operates on the network's global feature vector aggregated after the activation. For simple MLPs, the output directly represents the global feature vector. Networks that extract feature maps/sequences compute the global feature vector by taking the global average of the feature maps/sequences along the channels. Regarding some Transformer models that incorporate a class token, simply peel off the class token separately as the global feature vector while applying the corresponding constraint.

## 5 Experimental Study

**Datasets.** In the main experiments, we adopt five datasets, including MNIST (LeCun et al., 1998), CIFAR-10 (Krizhevsky et al., 2009), CIFAR-100 (Krizhevsky et al., 2009), ImageNet-100 (Deng et al., 2009), and ImageNet-1K (Deng et al., 2009), to verify the effectiveness of the proposed DNRT.

**Compared methods.** We conduct comparisons between DNRT and existing neural response mechanisms equipped with different mainstream activations (see §2.2), including Softplus (Dugas et al., 2000), ELU (Clevert et al., 2015), SELU (Klambauer et al., 2017), SiLU (Ramachandran et al., 2017), ReLU (Glorot et al., 2011), GELU (Hendrycks & Gimpel, 2016), and GDN (Ballé et al., 2015).

**Experimental settings.** The image size of MNIST and CIFAR-{10, 100} remains 28×28 and 32×32, while the images in ImageNet-{100, 1K} are uniformly scaled to 224×224. In the proposed ARR, the momentum $m$ for updating the moving mean is empirically set to 0.1, and the balanced parameter $\lambda$ varies depending on networks and datasets (see Appendix A.2). All experiments use the same data augmentations provided by timm (Wightman, 2019), AdamW optimizer with weight decay of 0.05, drop-path rate of 0.1, gradient clipping norm of 1.0, and cosine annealing learning rate scheduler with linear warm-up. Except for simple MLPs, which are trained for only 50 epochs from scratch, other networks are trained for 300 epochs from scratch. The automatic mixed precision is adopted to speed up the training. All other training settings, including batch size, learning rate, warm-up epochs, and so on, are kept identical throughout each set of comparative experiments. In the displayed tables, for example, "GELU" denotes the network with the GELU activation and the original response aggregation mechanism, and "DNRT" denotes the network with the proposed RAA and ARR.

Table 1: Top-1 accuracy (%) across the MNIST, CIFAR-10, and CIFAR-100 datasets using the proposed DNRT on a simple MLP with only one input layer, one hidden layer, and one output layer.

| Top-1 Acc / % | | Softplus | ELU | SELU | SiLU | ReLU | GELU | GDN | DNRT |
|---|---|---|---|---|---|---|---|---|---|
| MNIST | MLP | 97.3 | 97.5 | 97.6 | 97.0 | 97.1 | 97.5 | 97.5 | **98.0** |
| CIFAR-10 | MLP | 51.7 | 51.4 | 50.6 | 52.8 | 52.1 | 52.9 | 44.4 | **53.6** |
| CIFAR-100 | MLP | 24.8 | 24.7 | 24.4 | 26.1 | 25.4 | 25.6 | 19.8 | **26.1** |

Table 2: Top-1 accuracy (%) across the CIFAR-10, CIFAR-100, and ImageNet-100 datasets using the proposed DNRT on Vision Transformer (ViT) and its variants.

| Top-1 Acc / % | | Softplus | ELU | SELU | SiLU | ReLU | GELU | GDN | DNRT |
|---|---|---|---|---|---|---|---|---|---|
| | ViT-Tiny | 84.3 | 82.0 | 79.4 | 85.5 | 89.9 | 89.2 | 81.8 | **92.1** |
| | DeiT-Tiny | 84.7 | 81.4 | 79.9 | 86.6 | 89.6 | 89.2 | 83.0 | **92.4** |
| CIFAR-10 | CaiT-XXS | 82.5 | 80.7 | 78.4 | 86.6 | 89.4 | 88.7 | 80.0 | **91.6** |
| | PVT-Tiny | 90.6 | 89.3 | 85.4 | 92.5 | 93.0 | 92.8 | 82.8 | **94.3** |
| | TNT-Small | 88.3 | 85.4 | 83.7 | 90.5 | 90.9 | 91.2 | 85.1 | **92.5** |
| | ViT-Tiny | 62.4 | 60.0 | 57.5 | 65.5 | 65.7 | 65.4 | 59.4 | **71.4** |
| | DeiT-Tiny | 63.4 | 60.0 | 58.3 | 67.1 | 67.0 | 67.0 | 59.8 | **71.4** |
| CIFAR-100 | CaiT-XXS | 60.4 | 59.3 | 55.8 | 63.9 | 65.8 | 65.5 | 56.2 | **70.6** |
| | PVT-Tiny | 69.5 | 69.3 | 65.7 | 70.2 | 70.9 | 70.6 | 64.4 | **71.6** |
| | TNT-Small | 65.2 | 63.8 | 60.9 | 65.1 | 65.4 | 64.4 | 62.5 | **71.9** |
| | ViT-Tiny | 74.1 | 68.9 | 66.4 | 74.1 | 75.4 | 76.4 | 67.9 | **80.9** |
| | DeiT-Tiny | 75.3 | 69.4 | 67.0 | 75.1 | 75.6 | 74.6 | 66.3 | **81.1** |
| ImageNet-100 | CaiT-XXS | 70.9 | 69.1 | 65.9 | 76.1 | 76.0 | 76.7 | 69.5 | **80.4** |
| | PVT-Tiny | 79.5 | 77.1 | 76.1 | 79.5 | 81.9 | 81.4 | 75.8 | **84.1** |
| | TNT-Small | 78.9 | 79.3 | 76.4 | 77.6 | 79.9 | 77.2 | 76.9 | **82.3** |

Table 3: Top-1 accuracy (%) across the CIFAR-10, CIFAR-100, and ImageNet-100 datasets using the proposed DNRT on various CNN architectures.

| Top-1 Acc / % | | Softplus | ELU | SELU | SiLU | ReLU | GELU | GDN | DNRT |
|---|---|---|---|---|---|---|---|---|---|
| | AlexNet | 76.1 | 84.3 | 82.6 | 84.3 | 86.0 | 85.2 | 83.8 | **86.4** |
| | VGG-11 | 89.6 | 90.4 | 89.1 | 91.7 | 92.2 | 92.0 | 89.3 | **92.6** |
| CIFAR-10 | ResNet-18 | 94.0 | 93.9 | 93.0 | 95.2 | 95.0 | 95.2 | 82.3 | **95.5** |
| | MobileNet | **90.0** | 89.8 | 87.7 | 89.7 | 87.4 | 89.4 | 86.6 | **90.0** |
| | ShuffleNet | 90.2 | 90.2 | 87.8 | 90.8 | 89.4 | 90.9 | 87.6 | **91.2** |
| | AlexNet | 44.0 | 57.6 | 55.7 | 57.3 | 57.2 | 57.4 | 56.5 | **59.2** |
| | VGG-11 | 64.7 | 68.8 | 66.0 | 70.8 | 70.2 | 70.7 | 70.2 | **71.0** |
| CIFAR-100 | ResNet-18 | 75.6 | 75.5 | 74.7 | 75.7 | 75.7 | 75.8 | 71.7 | **76.5** |
| | MobileNet | 66.3 | **67.5** | 64.1 | 66.0 | 66.0 | 66.2 | 55.3 | 66.9 |
| | ShuffleNet | 68.0 | 68.2 | 63.8 | 68.0 | 66.3 | 68.2 | 57.3 | **68.7** |
| | AlexNet | 74.8 | 77.5 | 75.7 | 77.8 | 76.3 | 77.7 | 74.4 | **78.8** |
| | VGG-11 | 80.1 | 83.6 | 80.6 | 87.0 | 87.7 | 87.7 | 85.3 | **88.7** |
| ImageNet-100 | ResNet-18 | 85.4 | 84.9 | 83.9 | 85.7 | 84.9 | 85.9 | 80.2 | **86.8** |
| | MobileNet | 80.5 | 80.7 | 77.5 | 80.9 | 80.6 | 81.0 | 73.6 | **81.7** |
| | ShuffleNet | 83.0 | 82.3 | 79.4 | 82.9 | 81.6 | 82.6 | 75.3 | **83.4** |

## 5.1 DNRT ON MLP

As shown in Table 1, the proposed DNRT is firstly validated on the original ANN model, known as MLP (McClelland et al., 1987), due to its working mechanism being akin to the human brain's perception mechanism. A basic MLP compiled with one input layer, one hidden layer, and one output layer is built, with an activation applied after the hidden layer. DNRT can replace existing neural response mechanisms in MLP. The results indicate that the proposed DNRT yields the best performance on the classical MNIST, CIFAR-10, and CIFAR-100 datasets. It affirms the feasibility of incorporating the biological signal transmission and aggregation characteristics into ANNs.

Table 4: Generalization effect of the proposed DNRT across various tasks: node classification on the DGraph dataset and tackling highly imbalanced data on the Long-Tailed CIFAR-10 dataset.

| | Node classification | | Tackling highly imbalanced data ($\beta$=10, 50, 100) | | |
|---|---|---|---|---|---|
| AUC / % | GCN | GraphSAGE | Top-1 Acc / % | ViT-Tiny | ResNet-32 |
| ReLU | 72.5 | 75.1 | ReLU | 53.0, 50.0, 44.9 | 86.8, 83.5, 80.0 |
| GELU | 71.9 | 74.4 | GELU | 52.6, 49.3, 44.4 | 86.5, 83.3, 79.6 |
| DNRT | **74.9** | **76.7** | DNRT | **56.5, 53.6, 47.3** | **87.7, 84.1, 82.0** |

Table 5: Ablation study of the proposed DNRT on the CIFAR-100 dataset.

| Top-1 Acc / % | ReLU | GELU | RAA | ReLU + ARR | GELU + ARR | DNRT (RAA + ARR) |
|---|---|---|---|---|---|---|
| ViT-Tiny | 65.7 | 65.4 | 66.5 | 69.6 | 68.5 | **71.4** |
| ResNet-18 | 75.7 | 75.8 | 76.1 | 76.3 | 76.2 | **76.5** |

## 5.2 DNRT on ViTs

The popular Vision Transformer (ViT) and its variants incorporate multiple MLP structures. We incorporate the proposed DNRT mechanism into them. Table 2 shows the top-1 accuracy (%) of DNRT on CIFAR-10, CIFAR-100, and ImageNet-100 datasets across five different ViT architectures: ViT (Dosovitskiy et al., 2020), DeiT (Touvron et al., 2021a), CaiT (Touvron et al., 2021b), PVT (Wang et al., 2021) and TNT (Han et al., 2021). DNRT can replace existing neural response mechanisms in Transformer's MLPs, where RAA replaces the original activation, and ARR is performed on the class token aggregated after the activation. The results consistently illustrate that DNRT remarkably outperforms the baselines. Moreover, the results on ImageNet-1K can be found in Appendix A.3.

## 5.3 DNRT on CNNs

The proposed DNRT is also evaluated on various mainstream CNNs, including AlexNet (Krizhevsky et al., 2017), VGG (Simonyan & Zisserman, 2014), ResNet (He et al., 2016), MobileNet (Howard et al., 2017) and ShuffleNet(V2) (Ma et al., 2018), where RAA replaces the original activation, and ARR is applied to the globally pooled feature maps aggregated after the activation. The results in Table 3 highlight the versatility of DNRT in handling different CNN architectures.

## 5.4 Generalization to Other Tasks

Apart from the standard image classification task, the proposed DNRT can also perform various other tasks or domains, including non-computer vision tasks like GNN-based node classification, and some special tasks like tackling highly imbalanced data. The results are remarkable.

**Node classification.** We conduct additional experiments on the node classification task. The DGraph dataset (Huang et al., 2022) is employed to evaluate the proposed DNRT. GCN (Kipf & Welling, 2016) and GraphSAGE (Hamilton et al., 2017) are chosen GNN models, each comprising two layers with a 64-dimensional feature space. Table 4 (left side) shows the results.

**Tackling highly imbalanced data.** Additionally, the proposed DNRT is quite capable of addressing the negative impact brought by the highly imbalanced category distribution. A Long-Tailed CIFAR-10 dataset is generated with reduced training examples per class while the validation set remains unchanged. The imbalance rate $\beta$ denotes the ratio of the most to the least frequent category's sample size, i.e., $\beta = N_{max}/N_{min}$. The sample size of different classes decays exponentially. In this case, the primary loss item $\mathcal{L}_{task}$ employed in Eq. 7 is the "Balanced Softmax Loss (BSL)". The results for the imbalance rate $\beta$=10, 50, 100 are presented in Table 4 (right side).

## 5.5 Ablation Study

Table 5 presents the ablation results on the proposed DNRT. Both RAA and ARR contribute to enhancing the network, and their combination leads to further improvement in performance.

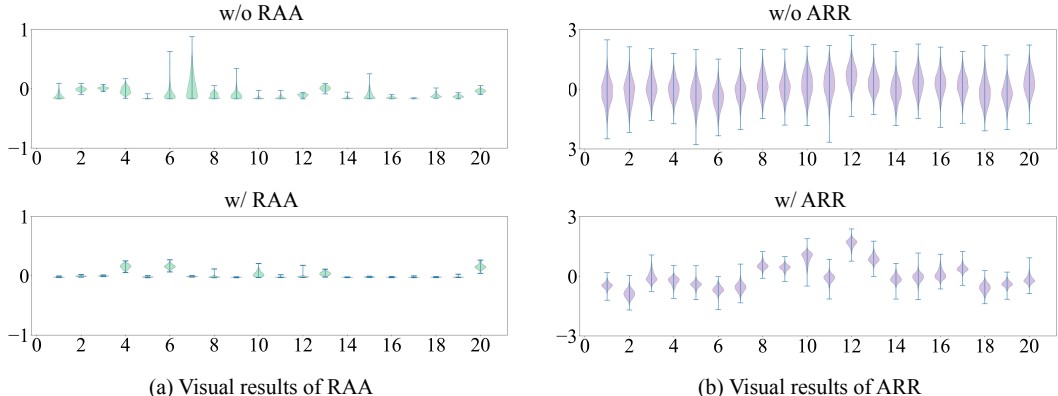

(a) Visual results of RAA          (b) Visual results of ARR

Figure 2: Response distributions on the class token in ViT's MLP block with and without the proposed DNRT on the CIFAR-10 dataset. (a) MLP's activation responses. (b) MLP's aggregated responses. The baseline is the network with the GELU activation (the default used in ViT) and the original response aggregation mechanism. The horizontal coordinate represents the element's (channel's) index within the feature vector, the vertical coordinate represents the response value, and the area depicts the response density. The figure displays the response distributions of the first 20 elements (channels) in each vector (the class token). For more results, please see Figure 4 in Appendix A.4.

## 5.6 NEURAL RESPONSE VISUALIZATION

Figure 2 displays the response distributions on the class token in ViT's MLP block with and without the proposed DNRT. As shown in Figure 2(a), when using a traditional activation mechanism with a static response condition, almost all channels exhibit a large amount of response since some irrelevant features that should not be activated are mistakenly activated. In contrast, RAA reduces the activation of irrelevant features and achieves a considerably sparser activation response, indicating the network's ability to self-regulate response conditions and extract key features from input data with higher precision. Figure 2(b) illustrates the effect of imposing the ARR on ViT's class token. Under the guidance of ARR, the MLP's aggregated response on the class token becomes more concentrated, enabling the network to emphasize task-related features, thus improving its generalization ability. These visual results are in line with our expectations, providing clear insights into the internal mechanism of DNRT and elucidating why DNRT can elevate the performance of ANNs. More visual results and discussions can be found in Figure 4 and Appendix A.4.

## 6 CONCLUSION

Inspired by the principles of biology and neuroscience, we propose a novel mechanism called Dynamic Neural Response Tuning (DNRT). Responsive-Adaptive Activation (RAA) and Aggregated Response Regularization (ARR) are devised to simulate the biological neuron's behaviors of transmitting and aggregating information, dynamically adjusting response conditions, and regularizing the gathered activated signals, respectively. The experiments demonstrate that the proposed DNRT is more effective than existing neural response mechanisms across various ANN architectures, including MLP, ViTs, and CNNs. Furthermore, DNRT showcases its superiority in diverse tasks and domains, encompassing image classification, non-computer vision tasks like node classification, and some special tasks like tackling highly imbalanced data. A more precise simulation of response characteristics in biological neural networks can further enhance ANNs. Overall, the findings of this paper have profound implications for the ANN's future. Moving forward, we are going to delve deeper into the analysis of both biological neural networks and representation-level patterns within ANNs to devise more advanced neural response tuning mechanisms.

### ACKNOWLEDGMENTS

This work is supported by National Natural Science Foundation of China (U20B2066) and Ningbo Natural Science Foundation (2022J182, 2023J281).

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

# A  APPENDIX

## A.1  COMPUTATIONAL COSTS

This section discusses the computational costs of the proposed DNRT in terms of trainable parameters. GPU overhead, computational complexity, and inference speed. An experiment is also conducted to validate that the primary performance improvement comes from the RAA intrinsic mechanism rather than simply the extra parameters.

Firstly, in the proposed DNRT, ARR is a training strategy that does not introduce any parameters. RAA needs negligible additional parameters to judge the relevance of input signals. Table 6 compares DNRT's trainable parameters and GPU overhead with those of the original networks. The networks are fed 224×224-pixel images with a batch size of 512, and the GPU occupancy is recorded. Notably, the activation mechanism used in original networks should be implemented manually as our RAA. Utilizing counterparts directly from pre-made libraries (like torch.nn) can result in unfair comparisons due to their high optimization at the low level. The experiment is conducted on an NVIDIA A100 (80G). For most ViTs and CNNs, the increase in trainable parameters ranges from 0.008% to 0.18%, which is negligible, and the GPU occupancy during both training and inference is also negligible. The extremely low computational costs indicate that DNRT has broad practical value.

Table 6: Trainable parameters (M) and GPU overhead (GiB) of the networks with the proposed DNRT compared to those of the original networks. "RAA" denotes the network with RAA only. "DNRT" denotes the network with both RAA and ARR.

| Network / Metric | Trainable Parameters / M | | | Training GPU Memory / GiB | | | Inference GPU Memory / GiB | | |
|---|---|---|---|---|---|---|---|---|---|
| | Original | RAA | DNRT | Original | RAA | DNRT | Original | RAA | DNRT |
| ViT-Tiny | 5.38 | 5.39 | 5.39 | 19.76 | 19.90 | 19.91 | 4.18 | 4.19 | 4.19 |
| DeiT-Tiny | 5.91 | 5.92 | 5.92 | 20.04 | 20.19 | 20.20 | 4.28 | 4.28 | 4.28 |
| CaiT-XXS | 11.96 | 11.98 | 11.98 | 47.67 | 47.82 | 47.84 | 3.86 | 3.86 | 3.86 |
| PVT-Tiny | 13.23 | 13.24 | 13.24 | 44.57 | 47.21 | 47.46 | 14.21 | 14.21 | 14.21 |
| TNT-Small | 23.76 | 23.77 | 23.77 | 69.39 | 69.69 | 71.15 | 7.84 | 7.85 | 7.85 |
| AlexNet | 61.10 | 61.11 | 61.11 | 7.87 | 8.42 | 8.42 | 3.99 | 4.36 | 4.36 |
| VGG-11 | 132.87 | 132.88 | 132.88 | 51.38 | 55.41 | 55.59 | 21.22 | 27.44 | 27.44 |
| ResNet-18 | 11.69 | 11.69 | 11.69 | 14.30 | 15.59 | 15.61 | 5.47 | 5.47 | 5.47 |
| MobileNet | 3.50 | 3.51 | 3.51 | 21.35 | 22.45 | 22.46 | 9.26 | 9.34 | 9.34 |
| ShuffleNet | 2.28 | 2.28 | 2.28 | 11.99 | 13.46 | 13.47 | 4.44 | 4.46 | 4.46 |

Moreover, the impact of DNRT on computational complexity and inference speed is also minimal. Here, we present numerical comparisons for the network's computational complexity, measured in "FLOPs" (floating-point operations), and inference speed, measured in "Latency" (the time required for model inference on a single image). The "Latency" is obtained, on average, from the model inferring 224×224-pixel images on an NVIDIA 3090. The results are shown in Table 7.

Table 7: Computational complexity (FLOPs / G) and inference speed (Latency / ms) of the networks with the proposed DNRT compared to those of the original networks.

| Network / Metric | ReLU (FLOPs) | ReLU (Latency) | DNRT (FLOPs) | DNRT (Latency) |
|---|---|---|---|---|
| ViT-Tiny | 1.078 G | 4.5 ms | 1.080 G | 4.6 ms |
| TNT-Small | 4.849 G | 10.1 ms | 4.856 G | 10.5 ms |

Additionally, concerning that the performance improvement brought by RAA may stem from the increased number of parameters itself, to verify the primary contributors to the performance improvement, we conduct another group of experiments where a pure extra layer of linearity (ex-L) is added into ViT-T's MLPs, with the same number of parameters as RAA. The comparison on the CIFAR-100 dataset is shown in Table 8. The performance on RAA outperforms the pure ex-L, which indicates the primary improvement stems from the RAA intrinsic mechanism (eliminating redundant/irrelevant signals) rather than simply the extra parameters.

Table 8: Verification of the primary contributor to the performance improvement on the CIFAR-100 dataset. The "ex-L" is a pure extra layer of linearity with the same number of parameters as RAA.

| Method | ReLU | GELU | ex-L + ReLU | ex-L + GELU | RAA |
|---|---|---|---|---|---|
| Trainable Parameters (with ViT-Tiny) | 5380132 | 5380132 | 5389360 | 5389360 | 5389360 |
| Top-1 Acc / % | 65.7 | 65.4 | 65.6 | 65.7 | **66.5** |

## A.2 HYPERPARAMETER SELECTION

**Momentum $m$ for updating the moving mean μ in Eq. 5.** In our experiments, the momentum $m$ is empirically set to 0.1 without too much consideration. Theoretically, a high value of $m$ risks making the mean value unstable, and conversely, a low value of $m$ smoothens the mean value update, but it may cause the mean value to lag behind. As illustrated in Figure 3(a), the network's performance is not sensitive to the $m$ between around 0.1 and 0.8, and extreme $m$ values lead to negative effects.

**Balanced parameter $\lambda$ for the ARR loss $\mathcal{L}_{arr}$ in Eq. 7.** The optimal balanced parameter $\lambda$ for $\mathcal{L}_{arr}$ is specific to individual tasks. The relation between $\lambda$ and the classification accuracy of training CIFAR-100 with ViT-Tiny is depicted in Figure 3(b), where the optimal $\lambda$ is roughly 20, and too large $\lambda$ results in negative side effects. For other trials, we found the optimal $\lambda$ to be around 10 when training CIFAR-10 with ViT-Tiny and TNT-Small, approximately 5 when training CIFAR-100 with CaiT-XXS, etc. Overall, the network's performance is not sensitive to $\lambda$ if $\lambda$ is not too large. Searching the optimal $\lambda$ for each task can be time-consuming, but one thing is for sure: small $\lambda$ values do not hurt accuracy. Hence, for the experiments on ImageNet-1K, we set the default $\lambda$ to 1.

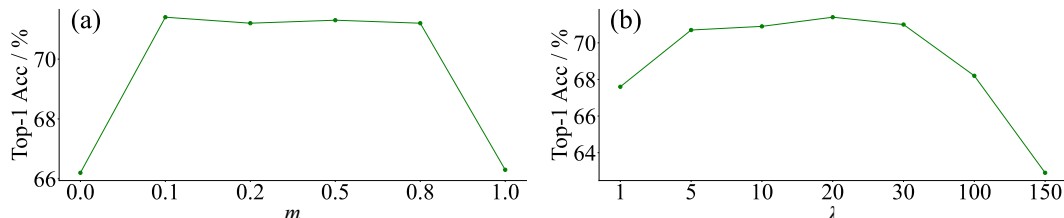

Figure 3: (a) Top-1 accuracy (%) w.r.t the momentum $m$ for updating the moving mean in Eq. 5 when training the CIFAR-100 dataset using the ViT-Tiny model. (b) Top-1 accuracy (%) w.r.t the balanced parameter $\lambda$ for $\mathcal{L}_{arr}$ in Eq. 7 when training the CIFAR-100 dataset using the ViT-Tiny model.

## A.3 EXPERIMENTS ON IMAGENET-1K

Table 9 presents the top-1 accuracy (%) achieved by the proposed DNRT on various ViT architectures when evaluated on the ImageNet-1K dataset. As with the results in Table 2, the ones on ImageNet-1K continue to highlight the consistent superiority of DNRT over the baseline methods, showcasing the scalability and robustness of the proposed approach.

Table 9: Top-1 accuracy (%) on the ImageNet-1K dataset using the proposed DNRT on Vision Transformer (ViT) and its variants.

| Top-1 Acc / % | | ViT-Tiny | DeiT-Tiny | CaiT-XXS | PVT-Tiny | TNT-Small |
|---|---|---|---|---|---|---|
| ImageNet-1K | ReLU | 70.9 | 73.2 | 74.0 | 73.7 | 73.4 |
| | GELU | 70.4 | 73.0 | 73.6 | 73.5 | 73.3 |
| | DNRT | **73.0** | **73.5** | **75.3** | **76.1** | **75.6** |

## A.4 MORE NEURAL RESPONSE VISUALIZATION

This section extends §5.6 - "Neural Response Visualization" of the main paper, providing more neural response visualization results in Figure 4. As shown in Figure 4(a), with the static response activation, the presented responses are active on almost every channel. Some irrelevant inputs that should not be activated are mistakenly activated, and of course, a few relevant inputs that should be activated might

be suppressed. After using the proposed RAA, the activation responses become significantly sparser, implying that most irrelevant features are suppressed, which can reduce the network's learning difficulty and improve the network's interpretability. Figure 4(b) shows the aggregated response distribution of ViT's MLP on the class token across different input categories. In the absence of ARR, the variance of aggregated responses on each channel of the class token is significantly high, making it challenging to differentiate class tokens for different objects visually. The use of ARR makes the response of each channel more focused, enabling the distinguishability of responses across different categories, thus facilitating the network's final decisions.

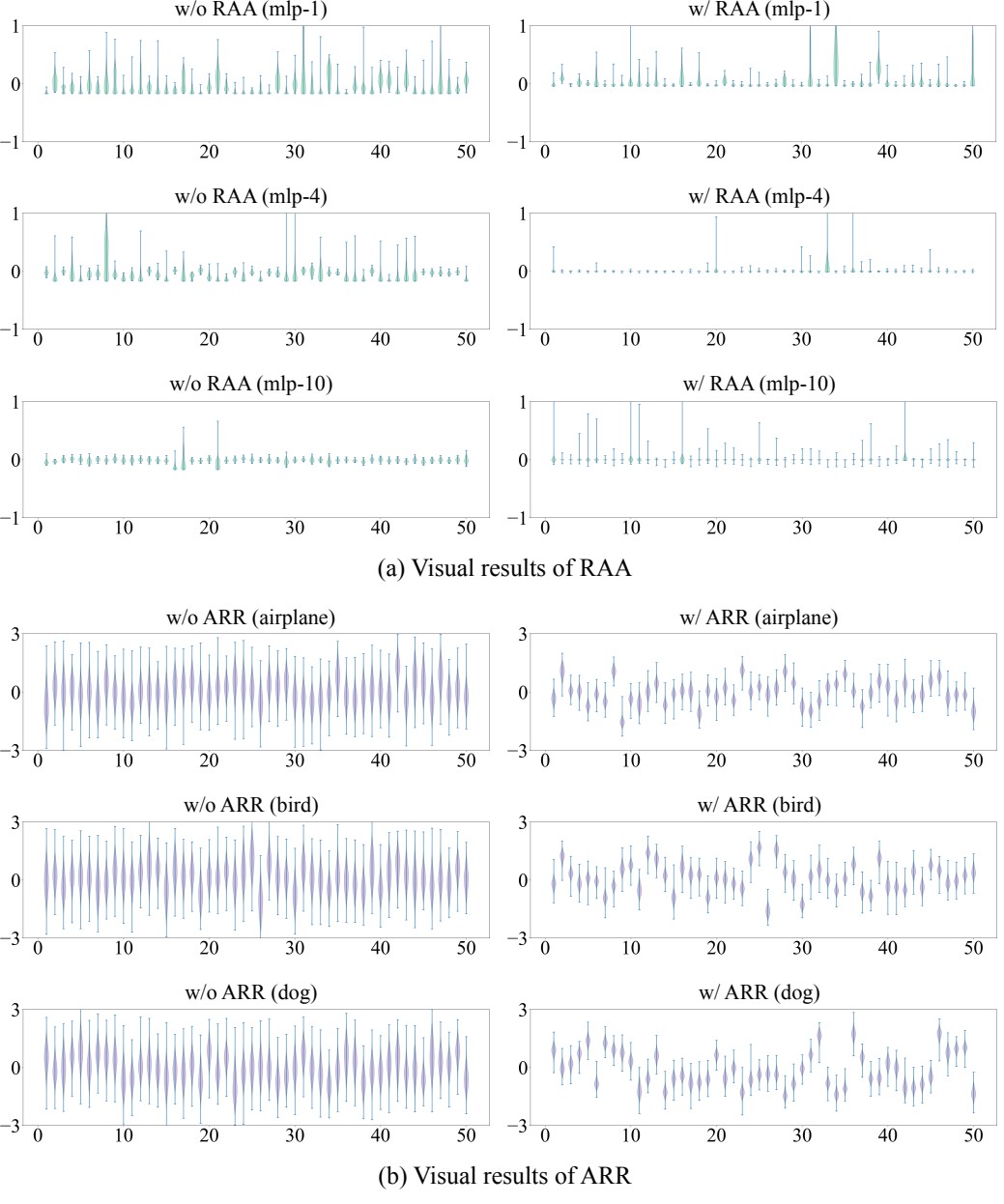

(a) Visual results of RAA

(b) Visual results of ARR

Figure 4: Response distributions on the class token in ViT's MLP block with and without the proposed DNRT on the CIFAR-10 dataset. (a) MLP's activation responses in ViT layer-1, 4, and 10. (b) MLP's aggregated responses to airplane, bird, and dog images. The baseline is the network with the GELU activation (the default used in ViT) and the original response aggregation mechanism. The horizontal coordinate represents the element's (channel's) index within the feature vector, the vertical coordinate represents the response value, and the area depicts the response density. The figure displays the response distributions of the first 50 elements (channels) in each vector (the class token).

## A.5 COMPARISON WITH SNNs

Leaky Integrate-and-Fire (LIF) (Maass, 1997) is used as the activation mechanism in Spiking Neural Networks (SNNs), which simulates the process of potential integration and output in biological neurons. When the potential accumulates to a certain threshold, it triggers a pulse output, after which the potential gradually returns to the baseline level through the leakage mechanism. The dynamic nature of LIF is reflected in two aspects:

*1. Time dependence of potential.* At each moment, there will be potential accumulation or release. The potential varies over time. However, it is currently still in the exploration stage, without demonstrating performance advantages over existing approaches.

*2. Learnable thresholds.* The original LIF in SNN (Maass, 1997) used a fixed threshold, and recently it has started supporting learnable thresholds (LIF-LT) (Wang et al., 2022), which shares similarities with our DNRT approach. However, LIF-LT simply determines the threshold by a scalar learnable parameter, which still lacks flexibility. In LIF-LT, neurons in the same layer share the same threshold and cannot adapt it based on the contextual input. As a result, it is unable to avoid irrelevant features incorrectly triggering activation responses. In contrast, DNRT judges the relevance of each input signal and adjusts the threshold accordingly for each signal, offering greater flexibility compared to LIF-LT's learnable threshold.

Additionally, Table 10 presents the numerical comparison on the CIFAR-10 dataset. In terms of performance, the mechanism of DNRT remains advantageous.

Table 10: Top-1 accuracy (%) of the proposed DNRT compared with SNN's Leaky Integrate-and-Fire (LIF) mechanisms on the CIFAR-10 dataset.

| Top-1 Acc / % | Softplus | ELU | SELU | SiLU | ReLU | GELU | GDN | LIF | LIF-LT | DNRT |
|---|---|---|---|---|---|---|---|---|---|---|
| ResNet-19 | 93.8 | 94.4 | 93.2 | 95.2 | 95.1 | 95.2 | 82.1 | 92.9 | 93.5 | **95.6** |

## A.6 OVERFITTING TEST

Deep ANNs are susceptible to suffering severe overfitting when the available data is highly insufficient, resulting in limited performance on the testing set. The proposed ARR mechanism is capable of alleviating this challenge. As previously mentioned, when the network's performance is poor, the aggregated response will exhibit a Gaussian distribution with high variance. ARR leverages historical response statistics and adds loss constraints to new responses, thereby achieving more focused response distributions within the same category. We evaluate the performance of the ViT-Tiny model with and without the proposed ARR on the CIFAR-10 dataset when using weak data augmentations only consisting of "random horizontal flipping" and "normalization". Figure 5 shows the loss and accuracy curves. Specifically, the loss curve on the testing set provides further evidence that ARR effectively overcomes overfitting. This is because the response constraint can effectively generalize the response clustering from the training set to the testing set.

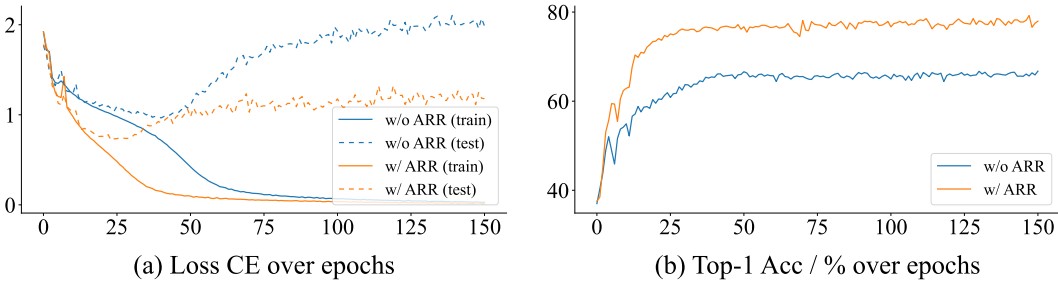

(a) Loss CE over epochs         (b) Top-1 Acc / % over epochs

Figure 5: The classification performance of ViT-Tiny with and without the proposed ARR on the CIFAR-10 dataset using weak data augmentations comprising only "random horizontal flipping" and "normalization". The model *without* ARR experiences severe overfitting, while the one *with* ARR overcomes it and yields massive performance gains.

