# OpenReview forum: "Dynamic Neural Response Tuning"
_ICLR.cc/2024/Conference — ICLR 2024 poster_

### Official Review · Reviewer_9nGD · 2023-10-31

**Soundness:** 2 fair
**Presentation:** 3 good
**Contribution:** 2 fair
**Rating:** 6
**Confidence:** 3

**Summary:**

The authors' rebuttal have essentially addressed my concerns, so I have raised the score of the paper to a weak accept.

=========================================================================================

This paper proposes a novel Dynamic Neural Response Tuning (DNRT) mechanism inspired by the dynamic response conditions of biological neurons. The DNRT mechanism comprises Response-Adaptive Activation (RAA) and Aggregated Response Regularization (ARR), which mimic the information transmission and aggregation behaviors of biological neurons. The authors demonstrate that the proposed DNRT can be applied to various mainstream network architectures and can achieve remarkable performance compared with existing response activation functions in multiple tasks and domains. The paper contributes to the simulation of dynamic biological neural response patterns and provides new concepts and techniques for constructing Artificial Neural Networks (ANNs).

**Strengths:**

Originality: The paper presents an innovative approach to mimicking the dynamic response conditions of biological neurons in ANNs, which is a significant departure from existing static activation functions.

Quality: The proposed DNRT mechanism is well-motivated and grounded in neuroscience research. The Response-Adaptive Activation (RAA) and Aggregated Response Regularization (ARR) components are clearly explained and justified.

Clarity: The paper is well-written and organized, with clear explanations of the DNRT mechanism, its components, and the motivation behind it. The figures and tables help illustrate the concepts and results effectively.

Significance: The DNRT mechanism has the potential to improve the performance of ANNs in various tasks and domains, as demonstrated by the extensive experiments conducted by the authors. The paper also provides valuable insights into neural information processing research.

**Weaknesses:**

Limited comparison with existing methods: While the paper demonstrates the performance of DNRT compared to existing activation functions, it would be helpful to see a more in-depth analysis and discussion of the differences between DNRT and other approaches, including spiking neural networks (SNNs). SNNs are based on neural dynamics and employ dynamic activation functions, which share similarities with the ideas presented in this paper.

Lack of generalization analysis: The paper claims that DNRT is applicable to various mainstream network architectures, but it would be beneficial to provide a more detailed analysis of how the proposed method generalizes to other tasks and domains beyond those presented in the experiments.

Potential implementation challenges: The paper does not discuss the possible challenges in implementing the proposed DNRT mechanism in real-world applications or the computational cost associated with its use.

**Questions:**

Can the authors provide more in-depth comparisons between the proposed DNRT mechanism and existing activation functions, discussing the advantages and disadvantages of each approach? Additionally, please include a comparison with spiking neural networks, as they also employ dynamic activation functions.

How does the DNRT mechanism generalize to other tasks and domains beyond those presented in the experiments? Are there any limitations or challenges in applying DNRT to different network architectures?

What are the potential implementation challenges and computational costs associated with using the DNRT mechanism in real-world applications? Are there any strategies to mitigate these challenges?

Considering the mainstream view that "computation is all you need," it is important to evaluate the additional computational costs and bandwidth bottlenecks introduced by DNRT. This is a concern as activation functions like Softplus and ELU are not widely used due to their lower computational efficiency compared to ReLU. Additionally, there are ongoing discussions about replacing the softmax layer in transformers with ReLU for improved computational efficiency. Could the authors provide an assessment of the impact of DNRT on model inference speed when replacing ReLU?

---

> ### Author Response · Authors · 2023-11-16
> **Author Response to Reviewer 9nGD (Part 1/2)**
>
> Thank you for dedicating your time and effort to reviewing our work. We will address your concerns and questions as follows:
>
> > **Q1:** (a) Can the authors provide more in-depth comparisons between the proposed DNRT mechanism and existing activation functions, discussing the advantages and disadvantages of each approach? (b) Additionally, please include a comparison with spiking neural networks, as they also employ dynamic activation functions.
>
> We appreciate your constructive suggestion.
>
> **(a)** The major disadvantage of existing activation functions in ANNs is their static nature, making them unable to identify and filter out all irrelevant signals. For instance, ReLU has a fixed activation threshold of 0, which allows it to quickly achieve sparse activation response with its simple and efficient design. However, due to its static nature, there will inevitably be some irrelevant stimuli that exceed the threshold and trigger activation responses in ReLU, hindering the key feature extraction. The same drawbacks exist in activations such as sigmoid, ELU, GELU, and those mentioned in Related Work §2.3. In contrast, the proposed RAA mimics the human brain's dynamic redundant feature filtering mechanism. It adjusts the threshold for each input signal based on the signal's relevance degree, thus avoiding the aforementioned flaw.
>
> **(b) Comparison with SNNs.** Leaky Integrate-and-Fire (LIF) is used as the activation mechanism in Spiking Neural Networks (SNNs), which simulates the process of potential integration and output in biological neurons. When the potential accumulates to a certain threshold, it triggers a pulse output, after which the potential gradually returns to the baseline level through the leakage mechanism.
>
> The dynamic nature of LIF reflects in two aspects:
>
> 1. *Time dependence of potential.* At each moment, there will be potential accumulation or release. The potential varies over time. However, it is currently still in the exploration stage, without demonstrating performance advantages over existing approaches.
>
> 2. *Learnable thresholds.* The original LIF in SNN used a fixed threshold, and recently it has started supporting learnable thresholds (LIF-LT) (Wang et al., NeurIPS 2022), which shares similarities with our DNRT approach. **However, LIF-LT simply determines the threshold by a scalar learnable parameter, which still lacks flexibility. In LIF-LT, neurons in the same layer share the same threshold and cannot adapt it based on the contextual input. As a result, it is unable to avoid irrelevant features incorrectly triggering activation responses. In contrast, our DNRT judges the relevance of each input signal and adjusts the threshold accordingly for each signal, offering greater flexibility compared to LIF-LT's learnable threshold.**
>
> Additionally, here is the numerical comparison on CIFAR-10. In terms of performance, the mechanism of DNRT remains advantageous.
>
> |Top-1 Acc / %|LIF|LIF-LT|DNRT|
> |-|-|-|-|
> |ResNet-19|92.9|93.5|**95.6**|
>
> Thank you once again for your valuable feedback. We will incorporate these discussions into the revised version.
>
> > **Q2:** (a) How does the DNRT mechanism generalize to other tasks and domains beyond those presented in the experiments? (b) Are there any limitations or challenges in applying DNRT to different network architectures?
>
> Thanks for your comment. Here is our explanation:
>
> **(a) Generalization Analysis.** Besides the image classification task, the proposed DNRT can perform various other tasks or domains, including but not limited to semantic segmentation, non-computer vision tasks like GNN-based node classification, and some special tasks like tackling highly imbalanced data. For numerical experimental details, please refer to **the \*initial\* submitted manuscript §5.5 (pp. 8) "Generalization to Other Tasks".**
>
> *(Note: We are not sure what "those presented in the experiments" in your comment refers to. If it refers to §5.5 of the initial paper, then we are willing to provide analysis on more other tasks and domains.)*
>
> **(b)** No, the proposed DNRT is adaptable to various ANN architectures including MLPs, CNNs, ViTs and GNNs. These ANN architectures are employed in a wide range of mainstream deep learning tasks. Table 1,2,3,5,6,7 in the paper can demonstrate the DNRT's broad applicability across different ANN architectures.

---

> ### Author Response · Authors · 2023-11-16
> **Author Response to Reviewer 9nGD (Part 2/2)**
>
> > **Q3:** What are the potential implementation challenges and computational costs associated with using the DNRT mechanism in real-world applications? Are there any strategies to mitigate these challenges?
>
> Thank you for your valuable feedback. We would like to address each of the points you've raised.
>
> **1. Implementation Challenges.** We have designed our DNRT to be adaptable to various ANN architectures including MLPs, CNNs, ViTs and GNNs. The code is modularized and can be easily integrated into existing systems without requiring extra special hardware support. Therefore, the implementation challenges of applying DNRT in real-world applications are relatively minimal.
>
> **2. Computational Costs**. Firstly, the proposed DNRT is designed with negligible additional **GPU overhead** during both training and inference. *Please refer to the \*initial\* submitted manuscript §A.4 (pp. 14~15) "Resource Demands" for numerical details*.
>
> Moreover, the impact of DNRT on **computational complexity** and **inference speed** is also minimal. Here we first present a numerical comparison on **computaional complexity**, measured in "FLOPs" (floating-point operations) as follows:
>
> |FLOPs|ReLU|DNRT|
> |-|-|-|
> |ViT-T|1.079 G|1.080 G|
> |TNT-S|4.849 G|4.856 G|
>
> We have also tested the network's **inference speed**, measured in "Latency" (the time required for model inference on a single image). The "Latency" is obtained, on average, from the model inferring images with 224×224 on a single GeForce RTX 3090 as follows:
>
> |Latency|ReLU|DNRT|
> |-|-|-|
> |ViT-T|4.5 ms|4.6 ms|
> |TNT-S|10.1 ms|10.5 ms|
>
> The result indicates that DNRT has relatively minimal impact on inference speed.
>
> Additionally, the gap in GPU overhead and inference speed can be further mitigated through low-level software and hardware optimization.
> &nbsp;
> Overall, DNRT is suitable for real-world applications while incurring minimal additional computational costs in terms of GPU overhead, computational complexity and inference speed, and we believe there is still a lot of space for low-level software and hardware optimization. We hope that these clarifications address your concerns.
>
> > **Q4:** Considering the mainstream view that "computation is all you need," it is important to evaluate the additional computational costs and bandwidth bottlenecks introduced by DNRT. This is a concern as activation functions like Softplus and ELU are not widely used due to their lower computational efficiency compared to ReLU. Additionally, there are ongoing discussions about replacing the softmax layer in transformers with ReLU for improved computational efficiency. Could the authors provide an assessment of the impact of DNRT on model inference speed when replacing ReLU?
>
> We appreciate your constructive suggestion. The impact of DNRT on model inference speed  when replacing ReLU is negligible. For numerical comparisons on inference speed, **please refer to our response to the previous question (*Q3*) above**. We will include these discussions in the revised version.
>
> &nbsp;
>
> We hope that our response can adequately address your concerns. If necessary, we are more than willing to provide further elaboration. Looking forward to your reply!

---

> ### Author Response · Authors · 2023-11-21
> **Looking forward to your response**
>
> Dear Reviewer 9nGD,
>
> Thank you once again for your time and effort in reviewing our paper. As the discussion period draws to a close (less than 24 hours), we would like to know if our previous rebuttal has addressed your concerns.
>
> The revised version (PDF) has been uploaded, in which all the updated parts have been highlighted in red. Please note that the appendix (pp.14~19) is located after the main paper. If you have further questions, please do not hesitate to let us know.
>
> We are eagerly looking forward to your response.
>
> Sincerely,
>
> Authors of Paper #6922

---

> ### Author Response · Authors · 2023-11-23
> **Thank you for your recognition and raising the score to a weak accept**
>
> Dear Reviewer 9nGD,
>
> We have noticed your update to the "Summary" section of your original review.
>
> We sincerely appreciate your recognition of our efforts in addressing your concerns and thank you for raising the score to a weak accept.
>
> Thank you once again for your time and effort in reviewing our paper.
>
> &nbsp;
>
> Best regards,
>
> Authors of Paper #6922

---

### Official Review · Reviewer_2aiC · 2023-10-31

**Soundness:** 2 fair
**Presentation:** 2 fair
**Contribution:** 2 fair
**Rating:** 3
**Confidence:** 3

**Summary:**

The authors propose a method using novel activation functions which allows neurons in an ANN to provide outputs that cluster around particular classes more closely, by analogy with some observed behaviors of biological neurons.

**Strengths:**

The program idea, to make individual neurons in an ANN responsive to context or input distribution or similar is a good one.

**Weaknesses:**

General review context: Please note that I am simply another researcher, with some overlap of expertise with the content of the paper. In some cases my comments may reflect points that I believe are incorrect or incomplete. In most cases, my comments reflect spots where an average well-intentioned reader might stumble for various reasons, reducing the potential impact of the paper. The issue may be the text, or my finite understanding, or a combination. These comments point to opportunities to clarify and smooth the text to better convey the intended story. I urge the authors to decide how or whether to address these comments. I regret that the tone can come out negative even for a paper I admire; it's a time-saving mechanism for which I apologize.

The text is somewhat confusing to me. Examples: 1. It does not clearly distinguish between ANNs and biological NNs (BNNs), which confuses the exposition; 2. The method is not (for me) explained clearly enough, so I did not come away with a clear sense of what it entailed.

I suspect that the literature grounding is incomplete (I'm not expert).

Reviewer limitation: I had trouble understanding this paper, the mechanics of the method, and why it would be beneficial. So perhaps I am not an optimal reviewer. Or perhaps this reflects the need for some degree of rewriting.

Notes to ICLR:

1. Please include line numbers in the template. They make reviewing much easier!

2. Please reformat the default bibliography style to make searching the bib easier! eg numbered, last name first, initials only except for last name.

**Questions:**

Bibliography: Perhaps reformat the bibliography for easier searching, eg numbered, last name first, initials only except for last name.

Abstract:

Is this first sentence an accurate summary of why ANNs have done so well? It seems there is much more to the matter.

"is achieved by triggering action potentials": Often. Neurotransmitters and non-spiking neurons can also have prominent roles.

"depending on neuron properties and": There is a broad literature on neurotransmitters and how they modulate neural behavior.

"dynamically adjusts": my sense from the paper is that the activation function is modified to weight inputs differently, but once trained the response does not change dynamically, ie its response properties are fixed. If this is correct, "dynamically" is not a correct word.

"ARR is devide": Maybe give some brief detail about this, to parallel the detail given about RAA is the sentence above.

1. Paragraph 1: Perhaps find more comprehensive reviews for ANN progress (eg Goodfellow, etc).

1. paragraph 3 "otherwise it suppresses": This is true for ReLU, but not for other activation functions eg tanh. Also, this would be a good place to cite some good literature reviews of flavors of activation functions.

"An input value ranked later among all inputs": Does this refer to time-dependent neural response? I did not see this addressed as a technique or effect in the paper.

"always adheres to a gaussian curve": Always? This framing is too simple. Also in this paragraph: the distinction between when referring to ANNs or BNNs is unclear.

"It is a a common ... final decision.": This strikes me as vague. Can the sentence be sharpened?

2.1 "surpasses the threshold": For spiking neurons (see above comment about non-spiking types and local spread of neurotransmitters).

2.2 "pivotal role in neural networks": Does this refer to ANN, BNN, or both?

Bottom of page 3, "Biological NNs...": This is an important observation. However, does the proposed method really emulate this, or does it modify the activation function to a new, static, form?

3.1, last sentence: While BNNs feature dynamically changing activation functions, does the proposed method do this (same as previous comment)? Perhaps this is really a statement about my lack of understanding of the proposed method.

3.2, first sentence: I believe this is too simple a description.

---

> ### Author Response · Authors · 2023-11-16
> **Author Response to Reviewer 2aiC (Part 1/3)**
>
> Thank you for your meticulous review of the text. We sincerely apologize for any wording that may have affected the rigor and clarity of the paper, and will address each of your points as follows:
>
> ### Bibliography
> > Perhaps reformat the bibliography for easier searching, eg numbered, last name first, initials only except for last name.
> >
> Thank you for your constructive comment. If ICLR allows, we will reformat it for easier searching.
>
> ### Abstract
>
> > *"Artificial Neural Networks (ANNs) have gained broad applications across various fields due to their brilliant architecture designs."*
> **Q1:** Is this first sentence an accurate summary of why ANNs have done so well? It seems there is much more to the matter.
>
> Sorry for the confusion. We will modify the sentence to: "Brilliant architecture design is an important factor for the widespread application of Artificial Neural Networks (ANNs)." The original sentence was intended to highlight the importance of ANN architecture design as the paper proposes a novel activation function, which is part of the ANN architecture.
>
> > *"The transmission of information in neurons is achieved by triggering action potentials that propagate through axons."*
> **Q2:** "is achieved by triggering action potentials": Often. Neurotransmitters and non-spiking neurons can also have prominent roles.
>
> Thank you for pointing it out. It should be "The transmission of information in neurons is *often* achieved by triggering action potentials that propagate through axons."
>
> > *"... the biological neuron's response conditions are dynamic, depending on neuron properties and the real-time environment."*
> **Q3:** "depending on neuron properties and": There is a broad literature on neurotransmitters and how they modulate neural behavior.
>
> Thank you for pointing it out. We have adjusted the expression to be more rigorous: "the biological neuron's response conditions are dynamic, depending on multiple factors such as neuron properties and the real-time environment."
>
> > *"RAA dynamically adjusts the response condition based on the strength and characteristics of the input signal."*
> **Q4:** "dynamically adjusts": my sense from the paper is that the activation function is modified to weight inputs differently, but once trained the response does not change dynamically, ie its response properties are fixed. If this is correct, "dynamically" is not a correct word.
>
> Thank you for your comment. In the view of contemporary science, the behavior of biological neurons is driven by complex biological mechanism/rules. Therefore, the dynamic behavior of certain neurons on activation threshold adjustment is also driven by specific rules. In other words, based on specific rules, neurons can possess the ability to behave dynamically. Similarly, in our proposed DNRT in ANNs, the behavior of DNRT is driven by learned complex rules. This behavior, which involves adjusting the activation threshold in response to real-time context, reflecting a high level of flexibility, can thus be considered dynamic.
>
> Additionally, for instance, the Capsule Network ("Dynamic Routing Between Capsules", Hinton et al. NIPS 2017) dynamically selects the feature vectors' propagation route, but this dynamic behavior is in fact driven by a pre-defined routing-by-agreement algorithm. DynamicViT (Rao et al. NeurIPS 2021) trained a lightweight prediction module "to determine which tokens to be pruned in a dynamic way". These methods driven by learned rules also use the word "dynamic", to some extent reflecting their general perspective on dynamic behavior.
>
> > *"ARR is devised to enhance the network’s ability to learn category specificity."*
> **Q5:** "ARR is devised": Maybe give some brief detail about this, to parallel the detail given about RAA is the sentence above.
>
> Thanks for your suggestion. We extend the sentence to: "ARR is devised to enhance the network’s ability to learn category specificity by imposing constraints to the network's response distribution."

---

> ### Author Response · Authors · 2023-11-16
> **Author Response to Reviewer 2aiC (Part 2/3)**
>
> ### §1 - Paragraph 1
>
> > **Q6:** Paragraph 1: Perhaps find more comprehensive reviews for ANN progress (eg Goodfellow, etc).
>
> Thanks for your constructive suggestion. In addition to the references already cited in the paper, we will include more reviews for ANN progress in the revised version including but not limited to:
>
> [1] Goodfellow et al., Deep learning. *MIT press*, 2016.
> [2] Pouyanfar et al., A survey on deep learning: algorithms, techniques, and applications. *ACM computing surveys*, 2018.
>
> ### §1 - Paragraph 3
>
> > **Q7:** Paragraph 3 "otherwise it suppresses": This is true for ReLU, but not for other activation functions eg tanh. Also, this would be a good place to cite some good literature reviews of flavors of activation functions.
>
> Thanks for your insightful comment. The sentence will be modified to "otherwise, *in most cases*, it suppresses ...". We will cite more reviews of flavors of activation functions in the revised version including but not limited to:
>
> [1] Sharma et al., Activation functions in neural networks. *Towards data sci*, 2017.
> [2] Nwankpa et al., Activation functions: comparison of trends in practice and research for deep learning. *ArXiv preprint arXiv:1811.03378*, 2018.
> [3] Dubey et al., Activation functions in deep learning: a comprehensive survey and benchmark. *Neurocomputing*, 2022.
>
> > **Q8:** "An input value ranked later among all inputs": Does this refer to time-dependent neural response? I did not see this addressed as a technique or effect in the paper.
>
> Sorry for the confusion. This sentence refers to the classical GELU activation function, which is not time-dependent like that in spiking neural networks. In short, GELU assumes that smaller inputs (ranks lower) is more likely to be suppressed, whose curve is similar to ReLU but better simulates reality. We compared GELU with the proposed DNRT in the experiment.
>
> ### §1 - Paragraph 5
> > *"Due to the presence of slight natural noise, the distribution always adheres to a Gaussian curve."*
> **Q9:** (a) "always adheres to a gaussian curve": Always? This framing is too simple. (b) Also in this paragraph: the distinction between when referring to ANNs or BNNs is unclear.
>
> (a) We apologize for our wording. A more rigorous expression would be: "Due to the presence of slight natural noise, the distribution *often approximately* adheres to a Gaussian curve." The Gaussian distribution is often employed as an approximate distribution for errors in the natural world, which can be explained by the [central limit theorem](https://en.wikipedia.org/wiki/Central_limit_theorem).
>
> (b) Sorry for the confusion. In this paragraph, the text before "In experiments with ANNs" is discussing BNNs, and the rest is about ANNs. We will modify the first sentence of this paragraph to: "... after the signals aggregate in the *biological* neural system ...".
>
> > *"It is a common assumption to consider a Gaussian distribution of errors between the model’s predicted value and the true value (the mean value). So, the high Gaussian variances indicate the specificity of the response is not strong enough. It makes the
> response less distinct between different categories, affecting the network’s final decision."*
> **Q10:** "It is a common ... final decision.": This strikes me as vague. Can the sentence be sharpened?
>
> Sorry for the confusion. **Based on the experimental observation in Fig. 1(b), 2(b) and 3(b), the ANN's aggregated response in each channel approximately adheres to Gaussian distribution**, which can also be explained by the central limit theorem, i.e., in the case of adding/averaging/aggregating independent and identically distributed (which is also a commonly used assumption in nature and machine learning) random variables, as the sample size increases, the distribution of this sum/mean/aggregation tends to be a Gaussian distribution even if the original variables themselves are not normally distributed. **The Gaussian distribution reflects the presence of error in ANN's responses, and the distribution variance reflects the magnitude of this error.** As shown in Fig. 3(b, left), the response is pretty noisy and severely overlaps between categories. **The high Gaussian variances indicate the specificity of the response is not strong enough, making it difficult to distinguish between different categories through their responses** compared with those in Fig. 3(b, right)**, posing a significant challenge for the ANN to make correct judgments on categories.**
>
> (The sharpened expression would be the connection of all sentences in **bold**.)

---

> ### Author Response · Authors · 2023-11-16
> **Author Response to Reviewer 2aiC (Part 3/3)**
>
> ### §2.1 - Paragraph 2
> > *"When the stimulus surpasses the threshold, an action potential that travels to the terminal is produced, ..."*
> **Q11:** "surpasses the threshold": For spiking neurons (see above comment about non-spiking types and local spread of neurotransmitters).
>
> Thank you for pointing it out. It should be "*For spiking neurons,* when the stimulus surpasses the threshold, an action potential that travels to the terminal is produced, ..."
>
> ### §2.2 - Paragraph 1
> > *"The activation function plays a pivotal role in neural networks ..."*
> **Q12:** "pivotal role in neural networks": Does this refer to ANN, BNN, or both?
>
> Sorry for the confusion. It refers to ANN since this section discusses the previous research on ANN's activation functions. We will modify the sentence to: "The activation function plays a pivotal role in *artificial* neural networks ..."*
>
> ### §2.2 - Paragraph 2
> > *"Biological neural networks comprise neurons responsible for distinct locations with different response thresholds that dynamically adjust to the environmental conditions."*
> **Q13:** Bottom of page 3, "Biological NNs...": This is an important observation. However, does the proposed method really emulate this, or does it modify the activation function to a new, static, form?
>
> We sincerely apologize for any potential confusion caused. **The proposed method is flexibly dynamic _(please refer to our explanation in Q4)_, and have the ability to change its activation threshold based on real-time context, which emulates the BNN's corresponding dynamic working pattern.** It's able to identify and filter out irrelevant signals by activation threshold adjustment to avoid irrelevant features incorrectly triggering activation responses. **Fig. 2(a) & 3(a) of the paper** intuitively demonstrate the effectiveness of the proposed method: With the static activation function, the presented responses are active on almost every channel. Some irrelevant inputs that should not be activated might be mistakenly activated. After using the proposed approach, the activation responses become significantly sparser, implying that most irrelevant features are suppressed. We hope our explanation can address any doubts or concerns you may have.
>
> ### §3.1
> > *"Biological neurons implement dynamic response conditions, which can help us improve existing activation functions."*
> **Q14:** last sentence: While BNNs feature dynamically changing activation functions, does the proposed method do this **(same as previous comment)**? Perhaps this is really a statement about my lack of understanding of the proposed method.
>
> We apologize once again for any potential confusion caused. **Please refer to our explanation in the previous question (*Q13*) above**, which we believe will adequately address your concern.
>
> ### §3.2
> > *"The information aggregated by biological neurons exhibits Gaussian distributions with a high level of category specificity/distinctiveness."*
> **Q15:** first sentence: I believe this is too simple a description.
>
> Thanks for your comment. We have made the description clearer: "For different categories of stimuli, the information aggregated by biological neurons exhibits significant differences, indicating the response specificity of neurons. Due to the presence of slight natural noise, the response often approximately adheres to a Gaussian distribution."
>
> &nbsp;
>
> Finally, thank you once again for your meticulous review. We will make necessary revisions to the wording issues mentioned in your comments. If you have any other unclear parts, we welcome your corrections and look forward to more discussions with you!

---

> ### Author Response · Authors · 2023-11-21
> **Looking forward to your response**
>
> Dear Reviewer 2aiC,
>
> Thank you once again for your time and effort in reviewing our paper. As the discussion period draws to a close (less than 24 hours), we would like to know if our previous rebuttal has addressed your concerns.
>
> The revised version (PDF) has been uploaded, in which all the updated parts have been highlighted in red. Please note that the appendix (pp.14~19) is located after the main paper. If you have further questions, please do not hesitate to let us know.
>
> We are eagerly looking forward to your response.
>
> Sincerely,
>
> Authors of Paper #6922

---

> > ### Comment · Reviewer_2aiC · 2023-11-22
> > **Response to authors' responses**
> >
> > My thanks to the authors for thorough responses to all the reviewers' questions.
> >
> > While I am still not sold on the paper, this is partly (or largely) because I do not understand the exposition (cf my original review). However, I see that two other reviewers found the exposition to be clear. So the problem in this case is likely on my end. Thus I am happy to defer the yea-nay decision to the other reviewers, and I can upgrade my review score as needed.
> >
> > Thank you!

---

> > > ### Author Response · Authors · 2023-11-23
> > > **Author Response to Reviewer 2aiC**
> > >
> > > Dear Reviewer 2aiC,
> > >
> > > Thank you for your time and feedback. Before the deadline of the author-reviewer discussion phase, we would like to share some recent updates with you:
> > >
> > > - **Reviewer 9nGD** have raised the score to a Weak Accept, who in the original review praised our paper as well-written and organized.
> > >
> > > - **Reviewer sidQ**, who rated our paper as an Accept (score 8), also appreciated the paper's clarity.
> > >
> > > - **Reviewer Hui7** did not raise any concerns on the paper's clarity.
> > >
> > > Please kindly review the latest updates from the other reviewers. We believe that their evaluations may help alleviate any concerns you had regarding the clarity of our paper. Your consideration is of great importance for us. Thank you!
> > >
> > > &nbsp;
> > >
> > > Best regards,
> > >
> > > Authors of Paper #6922

---

### Official Review · Reviewer_sidQ · 2023-11-01

**Soundness:** 4 excellent
**Presentation:** 3 good
**Contribution:** 4 excellent
**Rating:** 8
**Confidence:** 4

**Summary:**

The paper delves into the realm of Artificial Neural Networks and their foundational principles inspired by biological neural systems. A focal point is the discernible distinction between the static response conditions in traditional ANNs and the inherently dynamic nature of biological neurons. Recognizing this disparity, the authors propose the Dynamic Neural Response Tuning (DNRT) mechanism.

DNRT is a two-pronged approach comprising:

1. Response-Adaptive Activation (RAA): A novel activation function that dynamically modulates its response based on the characteristics and magnitude of the incoming signal. Unlike traditional activation functions, which maintain a static response, RAA adapts, mirroring the dynamic response conditions observed in biological neurons.

2. Aggregated Response Regularization (ARR): A mechanism designed to refine the aggregation of signals in the network. ARR aims to enhance the network's proficiency in discerning and classifying distinct categories, thereby boosting performance.

To validate the effectiveness of DNRT, the authors embarked on an extensive experimental evaluation across diverse architectures, including basic MLPs, ViTs and CNNs, Their findings suggest that DNRT not only integrates seamlessly across these architectures but also often outperforms traditional activation functions in terms of performance.

**Strengths:**

The paper introduces the DNRT mechanism, a fresh perspective in the realm of neural network activation functions. While traditional activation functions have been studied extensively, the idea of aligning ANNs with the dynamic response patterns of biological neurons offers a distinct and innovative approach. The two-pronged design of DNRT, with Response-Adaptive Activation (RAA) and Aggregated Response Regularization (ARR), presents a unique combination of ideas, each addressing specific challenges in neural network design. The inspiration drawn from the dynamic nature of biological neurons and the attempt to emulate that in ANNs is a commendable original endeavor.

The paper showcases a meticulous and rigorous experimental evaluation across various architectures, solidifying its claims about the efficacy of DNRT. The ablation studies provide deeper insights into the individual and combined contributions of the components of DNRT, enhancing the paper's technical quality.

The paper is well-structured, with a clear flow from motivation to proposal, followed by experimental validation. Concepts like RAA and ARR are explained with precision, allowing readers to grasp the core ideas effectively. While there's room for more intuitive visualizations, the current presentation ensures that readers familiar with the domain can understand the proposed mechanism and its implications.

Addressing the static nature of traditional activation functions and attempting to bridge the gap with the dynamic responses of biological neurons holds significant implications for the field of neural networks.
If DNRT can be consistently shown to enhance performance across various architectures, as suggested by the paper's results, it can lead to a paradigm shift in how activation functions are designed and implemented.

**Weaknesses:**

While the paper draws inspiration from the dynamic nature of biological neurons, a deeper exploration into the biological underpinnings could have been beneficial. More direct parallels between the DNRT mechanism and real-world neural behaviors would strengthen the paper's foundational claims. Consider integrating more biological studies or references that showcase the parallels between DNRT and actual biological neural behaviors, reinforcing the biological accuracy of the proposed mechanism.

The paper provides comprehensive experimental evaluations, but a detailed discussion on potential limitations or scenarios where DNRT might not be as effective is missing. Understanding these limitations can provide a more balanced view and guide future work. Dedicate a section to discuss potential limitations, challenges in broader applicability, or scenarios where traditional activation functions might still be preferable.

While the paper does address prior work in the domain, a more detailed comparison or discussion regarding how DNRT builds upon or differs from existing methods would offer readers a clearer perspective on its novelty. The related work section could be expanded to include more direct comparisons with existing activation functions or mechanisms, highlighting the unique contributions of DNRT.

The paper mentions the adaptability of RAA and hints at additional parameters, but a clear breakdown of how DNRT introduces and manages these parameters, especially in comparison to traditional activation functions, is lacking. Provide a subsection detailing the parameterization of RAA, including how these parameters are learned, their impact on network complexity, and potential challenges in optimization.

**Questions:**

Could the authors elaborate on the specific biological studies or evidence that influenced the design of DNRT? How closely does DNRT emulate the dynamic behaviors observed in biological neurons?

Are there specific architectures, tasks, or datasets where DNRT might not provide significant benefits or might even be counterproductive? What are the potential challenges or drawbacks associated with DNRT?

How are the parameters in the Response-Adaptive Activation (RAA) introduced and managed? How do these parameters impact network complexity, and are there challenges in optimizing these parameters during training?

Beyond the experimental setups, how does DNRT fare in real-world applications or larger, more complex datasets? Are there scalability concerns or other practical challenges?

---

> ### Author Response · Authors · 2023-11-16
> **Author Response to Reviewer sidQ (Part 1/2)**
>
> Thank you for your valuable feedback. We sincerely appreciate your positive acknowledgment of the contribution of our work, and we would like to address your suggestions for improvement as follows:
>
> > **Q1:** (a) Could the authors elaborate on the specific biological studies or evidence that influenced the design of DNRT? (b) How closely does DNRT emulate the dynamic behaviors observed in biological neurons?
>
> We appreciate your constructive feedback. Here, we provide more details on DNRT's biological plausibility and its ability to mimic the dynamic behaviors observed in biological neurons:
>
> **(a)** Regarding information transmission, dynamic threshold is a regulatory mechanism of biological neurons, which maintains neuronal homeostasis by adjusting the firing rate of action potentials and can be considered as an adaptation to membrane potential. This dynamic threshold mechanism has been widely observed in various nervous systems [1,2,3]. Moreover, the neuronal response threshold can also be influenced by both intrinsic and extrinsic factors such as medications [4] and nervous system ailments [5]. These biological evidences influenced the design of DNRT's RAA part.
>
> Regarding information aggregation, for different categories of stimuli, the information aggregated by biological neurons exhibits significant differences, indicating response specificity of neurons [6]. Due to the presence of slight natural noise, the response often approximately adheres to a Gaussian distribution [7,8]. These biological evidences influenced the design of DNRT's ARR part.
>
> **References**
> [1] Fontaine et al., Spike-threshold adaptation predicted by membrane potential dynamics in vivo. *PLOS computational biology*, 2014.
> [2] Azouz et al., Dynamic spike threshold reveals a mechanism for synaptic coincidence detection in cortical neurons in vivo. *In Proceedings of the national academy of sciences of the united states of america*, 2000.
> [3] Sun, Experience-dependent intrinsic plasticity in interneurons of barrel cortex layer IV. *Journal of neurophysiology*, 2009.
> [4] Arancio et al., Activity-dependent long-term enhancement of transmitter release by presynaptic 3’, 5’-cyclic gmp in cultured hippocampal neurons. *Nature*, 1995.
> [5] Knowland et al., istinct ventral pallidal neural populations mediate separate symptoms of depression. *Cell*, 2017.
> [6] Kreiman et al., Category-specific visual responses of single neurons in the human medial temporal lobe. *Nature*, 2000.
> [7] Fischer, A history of the central limit theorem: from classical to modern probability theory. *Springer science & business media*, 2010.
> [8] Averbeck et al., Neural correlations, population coding and computation. *Nature reviews neuroscience*, 2006.
>
> **(b)** Theoretically, the proposed RAA incorporates a dynamic activation threshold that adjusts in response to the real-time context. It has the ability to identify and filter out irrelevant signals by dynamic threshold adjustment to avoid irrelevant features incorrectly triggering activation responses, which highly emulates the BNN's corresponding dynamic threshold. The proposed ARR imposes constraints on the neural response distribution to achieve more focused response in each channel, enabling the distinguishability of responses across different categories, thus facilitating the ANN’s final decisions and attempting to approach the high discriminative ability of biological neural networks for categories. In practice, Fig. 2 & 3 of the paper intuitively showcase that DNRT behaves according to the biological plausibility we anticipated.

---

> ### Author Response · Authors · 2023-11-16
> **Author Response to Reviewer sidQ (Part 2/2)**
>
> > **Q2:** Are there specific architectures, tasks, or datasets where DNRT might not provide significant benefits or might even be counterproductive? What are the potential challenges or drawbacks associated with DNRT?
>
> Thank you for your valuable comment.
>
> 1\. *Architectures.* DNRT is adaptable to various ANN architectures including MLPs, CNNs, ViTs and GNNs. These ANN architectures are employed in a wide range of mainstream deep learning tasks. Table 1,2,3,5,6,7 in the paper can demonstrate the DNRT's broad applicability across different ANN architectures.
>
> 2\. *Tasks.* In certain scenarios e.g. the box regression part of object detection, the ARR mechanism of DNRT may not be applicable. This is because ARR is connected to categories, whereas box regression is not directly associated with categories. Despite this, the generalization ability of DNRT is already commendable, as it can perform various other tasks or domains, including image classification, semantic segmentation, non-computer vision tasks like GNN-based node classification, and some special tasks like tackling highly imbalanced data, whose generalization has been verified in experiments of the paper.
>
> 3\. *Datasets.* DNRT was not designed for a specific scenario from the beginning. Its internal mechanism is geared towards various types of data. The experiments in the paper were evaluated on general datasets such as CIFAR, ImageNet, ADE20K, etc., and achieved remarkable results. Therefore, we believe that in specific application scenarios, DNRT can be equally effective.
>
> > **Q3:** (a) How are the parameters in the Response-Adaptive Activation (RAA) introduced and managed? (b) How do these parameters impact network complexity, \(c) and are there challenges in optimizing these parameters during training?
>
> We appreciate your valuable comment.
>
> **(a)** RAA comprises *d+1* parameters: a weight vector $\mathbf{w}$ with length *d* and a scalar bias $b$. The code of RAA is modularized to manage these parameters.
>
> **(b)** The impact of these extra parameters on **computaional complexity** and **inference speed** is minimal. Here we first present a numerical comparison on computaional complexity, measured in "FLOPs" (floating-point operations) as follows:
>
> |FLOPs|ReLU|DNRT|
> |-|-|-|
> |ViT-T|1.079 G|1.080 G|
> |TNT-S|4.849 G|4.856 G|
>
> We have also tested the network's inference speed, measured in "Latency" (the time required for model inference on a single image). The "Latency" is obtained, on average, from the model inferring images with 224×224 on a single GeForce RTX 3090 as follows:
>
> |Latency|ReLU|DNRT|
> |-|-|-|
> |ViT-T|4.5 ms|4.6 ms|
> |TNT-S|10.1 ms|10.5 ms|
>
> The results indicates these extra parameters have minimal impact on computaional complexity and inference speed. Additionally, the gap in the inference speed can be further reduced through low-level software and hardware optimization.
>
> **\(c)** No, the parameters in RAA can be trained through backpropagation in an end-to-end manner.
>
> We hope that these clarifications could address your concerns.
>
> > **Q4:** Beyond the experimental setups, how does DNRT fare in real-world applications or larger, more complex datasets? Are there scalability concerns or other practical challenges?
>
> Thank you for your valuable feedback. We have designed our DNRT to be adaptable to various ANN architectures including MLPs, CNNs, ViTs and GNNs with negligible computational overhead. The code is modularized and can be easily integrated into existing systems without requiring extra special hardware support. Besides, the internal mechanism of DNRT can accommodate data from various categories and domains. Therefore, even when the datasets are larger and more complex, we believe the DNRT can adapt well to these diverse types of data through learning.
>
> &nbsp;
>
> Once again, we deeply appreciate your acknowledgment of our work and your valuable suggestions. We hope that our response to your suggestions will also receive your recognition.

---

> ### Author Response · Authors · 2023-11-21
> **Looking forward to your response**
>
> Dear Reviewer sidQ,
>
> Thank you once again for your time and effort in reviewing our paper. As the discussion period draws to a close (less than 24 hours), we would like to know if our previous rebuttal has addressed your concerns.
>
> The revised version (PDF) has been uploaded, in which all the updated parts have been highlighted in red. Please note that the appendix (pp.14~19) is located after the main paper. If you have further questions, please do not hesitate to let us know.
>
> We are eagerly looking forward to your response.
>
> Sincerely,
>
> Authors of Paper #6922

---

### Official Review · Reviewer_Hui7 · 2023-11-09

**Soundness:** 2 fair
**Presentation:** 2 fair
**Contribution:** 2 fair
**Rating:** 5
**Confidence:** 3

**Summary:**

The authors developed a Dynamic Neural Response Tuning (DNRT) mechanism inspired by the dynamic response conditions of biological neurons, addressing the limitations of static response conditions in existing activation functions. DNRT consists of Response-Adaptive Activation (RAA) and Aggregated Response Regularization (ARR), which mimic biological neurons' information transmission and aggregation behaviors, respectively. RAA adjusts the response condition based on input signal strength and characteristics, while ARR enhances the network's ability to learn category specificity. Experiments show that DNRT is interpretable, compatible with various network architectures, and outperforms existing activation functions in multiple tasks and domains.

**Strengths:**

The authors developed a Dynamic Neural Response Tuning (DNRT) mechanism inspired by the dynamic response conditions of biological neurons, addressing the limitations of static response conditions in existing activation functions. DNRT consists of Response-Adaptive Activation (RAA) and Aggregated Response Regularization (ARR), which mimic biological neurons' information transmission and aggregation behaviors, respectively. RAA adjusts the response condition based on input signal strength and characteristics, while ARR enhances the network's ability to learn category specificity. Experiments show that DNRT is interpretable, compatible with various network architectures, and outperforms existing activation functions in multiple tasks and domains.

**Weaknesses:**

1. The author have established a series of baseline experiments; however, the performance demonstrated in these experiments appears to be inferior to the results reported in previous literature.

2. RAA method introduces an additional parameter, W, which may compromise the fairness of comparisons with other activation functions.

3. The authors did not state clearly of the significance of the study. What is the inspiration of this study for neuroscience, or for the development of deep learning?

**Questions:**

1. In the presented experiments, it is crucial to disclose the number of parameters employed for each model. To ensure a fair comparison, the parameters for the baseline models must be comparable. Please provide this information in a clear and concise manner.

2. It is imperative that the chosen baseline models are well-established and representative of the current state of the art in deep learning. Notably, state-of-the-art deep learning methods have demonstrated the ability to achieve accuracy levels exceeding 85% on the ImageNet dataset. Please confirm that the selected baselines adhere to this standard.

3. Apart from the aspect of biological plausibility, it is essential to elucidate the motivations behind the development of RAA and AAR for the deep learning community. Please provide a comprehensive explanation of the underlying inspirations.

---

> ### Author Response · Authors · 2023-11-16
> **Author Response to Reviewer Hui7 (Part 1/2)**
>
> Thank you for your thoughtful review of our manuscript and your insightful feedback. We have carefully considered your comments and would like to address each point.
>
> > **Q1:** In the presented experiments, it is crucial to disclose the number of parameters employed for each model. To ensure a fair comparison, the parameters for the baseline models must be comparable. Please provide this information in a clear and concise manner.
>
> Thank you for your valuable feedback. Firstly, here is the quantitative comparison of the number of parameters employed for each model **(given in the \*original\* paper submission  §5.7 (pp. 9) & §A.4 (pp. 14~15) "Resource Demands")**.
>
> |Network|#Params of Baseline|#Params of Baseline with RAA only|#Params of Baseline with DNRT (RAA+ARR)|
> |-|-|-|-|
> |MLP|0.62 M|0.62 M|0.62 M|
> |ViT-T|5.75 M|5.76 M|5.76 M|
> |DeiT-T|5.91 M|5.92 M|5.92 M|
> |CaiT-XXS|11.96 M|11.98 M|11.98 M|
> |PVT-T|13.23 M|13.24 M|13.24 M|
> |TNT-S|23.76 M|23.77 M|23.77 M|
> |AlexNet|61.10 M|61.11 M|61.11 M|
> |VGG-11|132.87 M|132.88 M|132.88 M|
> |ResNet-18|11.69 M|11.69 M|11.69 M|
> |MobileNet|3.50 M|3.51 M|3.51 M|
> |ShuffleNetV2|2.28 M|2.28 M|2.28 M|
> |Segmenter|6.94 M|6.95 M|6.95 M|
> |GCN|7.84e-3 M|7.91e-3 M|7.91e-3 M|
> |GraphSAGE|1.55e-2 M|1.56e-2 M|1.56e-2 M|
> |ResNet-32|20.62 M|20.63 M|20.63 M|
>
> In the proposed DNRT, ARR does not introduce any parameters. A single RAA comprises only *d+1* parameters: a weight vector $\mathbf{w}$ with length *d* and a scalar bias $b$. The parameter growth rate of the models discussed in the paper ranges from 0.008% to 0.16%, which is negligible.
>
> **Regarding the concern on comparison fairness,** i.e., whether the RAA's introduced parameters are the primary contributor to the performance improvement, we give another experiment where an extra layer of linearity (ex-L) is added into ViT-T's MLPs, with the same parameter count as RAA. The comparison on CIFAR-100 is shown as follows **(also given in the \*original\* paper submission §5.7 (pp. 9) & §A.4 (pp. 14~15) "Resource Demands")**.
>
> |Method|ReLU|GELU|ex-L + ReLU|ex-L + GELU|RAA|
> |-|-|-|-|-|-|
> |#Params (with ViT-T)|5754472|5754472|5763700|5763700|5763700|
> |Top-1 Acc / %|65.7|64.0|66.0|64.2|**66.7**|
>
> The performance on RAA outperforms the pure ex-L, which indicates the primary improvement stems from the RAA intrinsic mechanism (eliminating redundant/irrelevant signals) rather than simply the extra parameters.
>
> Finally, we are open to making necessary revisions to include the parameter counts in the main tables.
>
> > **Q2:** It is imperative that the chosen baseline models are well-established and representative of the current state of the art in deep learning. Notably, state-of-the-art deep learning methods have demonstrated the ability to achieve accuracy levels exceeding 85% on the ImageNet dataset. Please confirm that the selected baselines adhere to this standard.
>
> We appreciate your constructive feedback. The baseline models in our paper are taken from [`timm`](https://github.com/huggingface/pytorch-image-models) and [`torchvision`](https://github.com/pytorch/vision/tree/main/torchvision) libraries, and the training settings on ImageNet-1K are provided by the [`timm`](https://github.com/huggingface/pytorch-image-models). We apologize for using a smaller batch size and less GPUs than previous literature during training due to the limited computing resources we had, which may result in the difference between the results we obtained and those reported in previous literature. However, in principle, we have ensured that each experiment in a set of comparative studies used the same batch size and other public training settings, allowing the results to fairly demonstrate that our approach can bring improvements across different model architectures.
>
> Moreover, we agree that the chosen baseline models need to represent the current state of the art in deep learning. Therefore, we employ a state-of-the-art model architecture `MaxViT-B` with the input image resolution of 384×384 and apply the proposed DNRT on it, the experimental results on ImageNet-1K are as follows:
>
> |Top-1 Acc / %|ReLU|GELU|DNRT|
> |-|-|-|-|
> |MaxViT-B|86.0|85.9|**86.5**|
>
> which demonstrate the effectiveness of the proposed DNRT on larger and more advanced models. We will incorporate this information to enhance the overall quality of the manuscript.

---

> ### Author Response · Authors · 2023-11-16
> **Author Response to Reviewer Hui7 (Part 2/2)**
>
> > **Q3:** Apart from the aspect of biological plausibility, it is essential to elucidate the motivations behind the development of RAA and AAR for the deep learning community. Please provide a comprehensive explanation of the underlying inspirations.
>
> We appreciate your suggestion which can help enhance the depth of our paper. Below, we provide a comprehensive explanation:
>
> ANNs fundamentally aim to replicate the operational principles of the human brain. Nevertheless, **imperfections within such artificial creations remain a constant, inviting us to identify and address these issues**. Biological plausibility (mimicking biological information transmission and aggregation) is one consideration in the design of DNRT(RAA+ARR). Another crucial aspect is that **deficiencies in ANNs concerning information transmission and aggregation do exist**, so the primary objective of DNRT is to **address the corresponding observed deficiencies of ANNs**, and no such targeted alternatives currently exist, which makes the design necessary, and it holds great promise in advancing the field and tackling future challenges for deep learning community.
>
> **Insights of RAA.** Existing activations in ANNs lack in-depth insights and often encounter challenges in achieving desired activation sparsity due to their static nature, as shown in the visual results of §5.6 & §A.3. The proposed RAA addresses this limitation by dynamically adjusting the activation threshold based on the relevance of each input signal. Sparser activation response helps the data fitting since only crucial features get activated while redundant/irrelevant features being suppressed.
>
> **Insights of ARR.** The aggregated response in ANNs tend to exhibit Gaussian distributions with high variances on each channel when stimulated with specific categories, as shown in the visual results of §5.6 & §A.3. The high variances suggests that the network's response to each category has large errors and is not discriminative enough. The proposed ARR addresses this flaw by imposing constraints to the ANN's response distribution, leading to improved category specificity and decision-making.
>
> In conclusion, apart from the aspect of biological plausibility, the motivation behind the development of DNRT(RAA+ARR) stems from the desire to overcome limitations in ANNs concerning information transmission and aggregation by replicating the complex operational principles of the human brain, helping enchance performance across various deep learning mdoels and tasks. As the deep learning community continues to evolve, the integration of DNRT offers a promising avenue for advancing the field and addressing the challenges ahead.
>
> &nbsp;
>
> We hope these responses effectively address your concerns. If you require any additional information or have further questions, please do not hesitate to let us know. We would be more than willing to engage in further discussion with you!

---

> ### Author Response · Authors · 2023-11-21
> **Looking forward to your response**
>
> Dear Reviewer Hui7,
>
> Thank you once again for your time and effort in reviewing our paper. As the discussion period draws to a close (less than 24 hours), we would like to know if our previous rebuttal has addressed your concerns.
>
> The revised version (PDF) has been uploaded, in which all the updated parts have been highlighted in red. Please note that the appendix (pp.14~19) is located after the main paper. If you have further questions, please do not hesitate to let us know.
>
> We are eagerly looking forward to your response.
>
> Sincerely,
>
> Authors of Paper #6922

---

> ### Comment · Reviewer_Hui7 · 2023-11-23
> **Thank you for your comment**
>
> I have carefully read the authors' rebuttal and while I acknowledge their efforts to address the concerns raised. The authors provided a full table of parameters, and showed that number of parameters are comparable. However, I remain unconvinced that the contributions presented in the paper are significant enough to warrant a change in my evaluation. Therefore, I will maintain my original score for the paper.

---

> > ### Author Response · Authors · 2023-11-23
> > **Author Response to Reviewer Hui7**
> >
> > Dear Reviewer Hui7,
> >
> > &nbsp;
> >
> > Thank you for your feedback. We would like to provide more explanations.
> >
> > ### Motivation
> > The proposed DNRT is inspired by the patterns of biological neuronal information transmission and aggregation, aiming to address the corresponding observed limitations of ANNs on the representation level. DNRT introduces dynamic characteristics to address the static nature of traditional activation functions and it attempts to bridge the gap with the high category specificity of biological neural responses by impose constraints on the response distribution. This approach is of great significance to the area of deep neural networks.
> >
> > ### Applicability
> > - *Architectures.* DNRT is adaptable to various ANN architectures including MLPs, CNNs, ViTs and GNNs. These ANN architectures are employed in a wide range of mainstream deep learning tasks. Table 1,2,3,5,6,7 in the paper can demonstrate the DNRT's broad applicability across different ANN architectures.
> >
> > - *Tasks.* DNRT can perform various other tasks or domains, including image classification, semantic segmentation, non-computer vision tasks like GNN-based node classification, and some special tasks like tackling highly imbalanced data, whose generalization has been verified in experiments of the paper §5.5.
> >
> > - *Datasets.* DNRT was not designed for a specific scenario from the beginning. Its internal mechanism is geared towards various types of data. The experiments in the paper were evaluated on general datasets such as CIFAR, ImageNet, ADE20K, etc., and achieved remarkable results. Therefore, we believe that in specific application scenarios, DNRT can be equally effective.
> >
> > In conclusion, the motivation/insight of DNRT is clear. DNRT's objective is to employ biological principles to address the drawbacks in information transmission and aggregation at the representation level of ANNs. DNRT has strong versatility and generalization, and can achieve significant performance improvement in various ANN architectures, tasks, and datasets. Therefore, we believe that DNRT's contribution to the neural information processing research as well as deep learning community is evident.
> >
> > &nbsp;
> >
> > Additionally, could you please provide more details regarding the specific areas in deep learning where our work lacks contribution? This information will help us to better understand the specific aspects that need improvement. Thank you!
> >
> > &nbsp;
> >
> > Best regards,
> >
> > Paper #6922 authors

---

### Author Response · Authors · 2023-11-22
**Author Response to All Reviewers: We are looking forward to receiving your responses**

Dear all reviewers,

Thank you once again for your time and effort in reviewing our paper.

As the discussion period draws to a close (less than 24 hours), we would like to know if our rebuttals have addressed your concerns.

We are eagerly looking forward to receiving your responses, as they hold great importance for us. Thank you for your attention.

&nbsp;

Sincerely,

Paper #6922 authors

---

### Meta-Review · Area_Chair_FQ1B · 2023-12-11

**Metareview:**

The paper introduces two mechanisms (jointly referred to as dynamic neural response tuning) that mimic some of the dynamic response adaptation mechanisms of biological neural networks. The first mechanism (RAA) is applying a learnable offset to the activation function that depends on the input strength. The second mechanism (ARR) aligns the neural responses to be similar for test examples from the same input category via an additional learning objective. Performance comparisons using different datasets for a range of feed-forward ANNs are shown, demonstrating a general overall performance benefits when applying these mechanisms.

The general idea of incorporating adaptive learning and response regularization mechanisms from biological neural networks  is commendable and was appreciated by the reviewers. The work proposes mechanisms that are relatively simple but effective. The extensive empirical verification demonstrates that the mechanisms can be incorporated in a wide range of network architectures and can be trained on various datasets. The enthusiasm for the work is damped by questions about it significance for the ANN community given the rather marginal improvements, concerns about the fairness of comparison given the additional degrees of freedom the mechanisms provide, and the lack of a thorough discussion about the limitations and drawbacks of implementing these mechanisms. As such the reviewers' scores were borderline, with the two extrema coming from reviewers that did not generally disagree but simply weighted the pro/cons differently according to their preferences.

The paper is borderline as it introduces on one hand an interesting new way to incorporate into ANNs some of the adaptive ideas of information transmission that make biological neural network so robust and efficient, but on the other hand fails to clearly address the limitations and drawbacks these mechanism practically impose. Is learning robust? What are the additional costs in maintaining the response histories for each input category during training (ARR)? Learning efficiency, transfer (when introducing new categories?

**Justification For Why Not Higher Score:**

The paper and the following discussion does/did not clearly discuss the limitations of the proposed mechanisms. There is no free lunch and the heavily supervised nature specifically of the ARR mechanisms (keeping track of input category specific response histories) is likely to have overall negative implications for learning efficiency, transfer, and robustness.

**Justification For Why Not Lower Score:**

It is very interesting to see potential improvements in ANN architectures and learning procedures that are directly inspired by biological NNs.

---

### Decision · Program_Chairs · 2024-01-16

Accept (poster)